# The human gut chemical landscape predicts microbe-mediated biotransformation of foods and drugs

**Leah Guthrie[1], Sarah Wolfson[1], Libusha Kelly[1,2]***

[1]Department of Systems and Computational Biology, Albert Einstein College of Medicine, New York, United States; [2]Department of Microbiology and Immunology, Albert Einstein College of Medicine, New York, United States

**Abstract** Microbes are nature's chemists, capable of producing and metabolizing a diverse array of compounds. In the human gut, microbial biochemistry can be beneficial, for example vitamin production and complex carbohydrate breakdown; or detrimental, such as the reactivation of an inactive drug metabolite leading to patient toxicity. Identifying clinically relevant microbiome metabolism requires linking microbial biochemistry and ecology with patient outcomes. Here we present MicrobeFDT, a resource which clusters chemically similar drug and food compounds and links these compounds to microbial enzymes and known toxicities. We demonstrate that compound structural similarity can serve as a proxy for toxicity, enzyme sharing, and coarse-grained functional similarity. MicrobeFDT allows users to flexibly interrogate microbial metabolism, compounds of interest, and toxicity profiles to generate novel hypotheses of microbe-diet-drug-phenotype interactions that influence patient outcomes. We validate one such hypothesis experimentally, using MicrobeFDT to reveal unrecognized gut microbiome metabolism of the ovarian cancer drug altretamine.

DOI: https://doi.org/10.7554/eLife.42866.001

***For correspondence:**
libusha.kelly@einstein.yu.edu

**Competing interests:** The authors declare that no competing interests exist.

## Introduction

Complex gut microbiome phenotypes shape human nutrition (*Martens et al., 2014*; *Sonnenburg et al., 2016*; *Bretin et al., 2018*), therapeutic drug responses (*Guthrie et al., 2017*; *Haiser et al., 2013*; *Koppel et al., 2017*) and disease susceptibility (*Koeth et al., 2013*). Multi'omic studies suggest that the human gut microbiota can be discretized at the resolution of microbial enzymes (*Guthrie et al., 2017*; *Tang and Hazen, 2014*), species (*Haiser et al., 2013*; *Haiser et al., 2014*), guilds (*Joossens et al., 2011*; *Wu et al., 2013*) or metabolites (*Clayton et al., 2009*) to characterize a range of human health and disease states. Gut microbial mediated biochemical transformations have consequences for drug treatment efficacy (*Koppel et al., 2017*; *Spanogiannopoulos et al., 2016*; *Alexander et al., 2017*; *Wilson and Nicholson, 2017*) and the etiology of inflammatory gastrointestinal diseases (*Tilg et al., 2018*; *Arthur et al., 2014*; *Belcheva et al., 2014*; *Brennan and Garrett, 2016*), however despite many examples there exist few unifying principles that govern microbiome impacts on human health.

Some microbiome/drug interactions have been characterized in detail. For example, the inactivation and decreased bioavailability of digoxin, a cardiac glycoside inhibitor, is linked to *cgr* operon expression levels in a single species, *E. lenta* (*Haiser et al., 2013*). Microbial β-glucuronidases mediate the reactivation of the key therapeutic metabolite of irinotecan, a chemotherapeutic prodrug used in the treatment of colorectal cancer, causing toxicity in some patients (*Guthrie et al., 2017*; *Wallace et al., 2010*). Notably, diet-derived compounds that are conjugated to glucuronic acid in

**eLife digest** Microbes in the human gut can play helpful roles by producing vitamins or breaking down complex carbohydrates. Collectively, gut microbes carry out these roles using a large toolkit of enzymes that catalyze a diverse range of chemical reactions, some of which cannot be carried out by human enzymes. However, these microbial enzymes can also cause harm if they alter drugs in a way that makes them toxic or prevents them from working. Little is known about which microbial enzymes interact with which foods and drugs, or how these interactions affect human health.

Guthrie et al. have now developed and tested a tool called MicrobeFDT that can help researchers to understand these complex interactions. In MicrobeFDT, 10,000 compounds produced by the human body or found in food or drugs are grouped based on their structure. Compounds are linked to the microbial enzymes that interact with them and drugs are annotated with information on known toxicities. The result is a network where compounds with similar structure are linked to each other.

If a microbial enzyme interacts with one compound in a group, it may interact with related compounds as well, potentially causing similar effects on human health. The network makes it easier for researchers to work out which compounds are affected by particular gut microbes. For example, MicrobeFDT suggested how gut microbes might alter the structure of an ovarian cancer drug called altretamine, which can cause diarrhea and kidney damage as side effects. Experiments confirmed that the predicted structural change does occur in human feces.

MicrobeFDT may increase how quickly researchers can assess harmful interactions between gut microbes, food, and drugs. It also may help them to develop new strategies to improve human health based on how microbial enzymes interact with food and drugs.

DOI: https://doi.org/10.7554/eLife.42866.002

the human liver and excreted via the biliary route into the GI tract are known substrates for microbial β-glucuronidases (*O'Leary et al., 2003*; *Sakurama et al., 2014*; *Maathuis et al., 2012*).

Many other gastrointestinally-routed drugs share overlapping chemical properties with diet-derived compounds. We understand in detail species-specific metabolism of some discrete chemical structures in dietary compounds, particularly polysaccharides (*Martens et al., 2008*); however we know little about the potential spectrum of drug metabolism by the microbiome.

Beyond the role of the microbiome in therapeutic drug treatment efficacy and polysaccharide metabolism, we have some mechanistic insight into how microbial metabolism contributes to host immunity. Microbial enzymes mediate the conversion of tryptophan into indole (*Sasaki-Imamura et al., 2010*) and indole derivatives (*Arora and Bae, 2014*) that shape human host immune responses (*Levy et al., 2017*; *Blacher et al., 2017*). Microbe produced indole 3-aldehyde functions as an activating ligand for human host aryl hydrocarbon receptors which are expressed by immune cells (*Zelante et al., 2013*). Indole binding induces IL-22 secretion by innate lymphoid cells, promoting the secretion of antimicrobial peptides that protects the host from pathogenic infection by *Candida albicans* (*Zelante et al., 2013*). Microbial production of short chain fatty acids (SCFAs) from dietary fiber also shapes host immunity, contributing to both innate and adaptive immune system functions (*Fukuda et al., 2011*; *Donohoe et al., 2011*; *Smith et al., 2013*).

Host-microbe interactions and phenotypes, ranging from host drug response to host immune response, are thus intimately connected to gut chemical signaling. Beyond these few well understood examples lie a vast space of uncharacterized microbe-drug-diet-phenotype interactions. We propose three key requirements to characterize the dynamics of the gut chemical space and its impact on health. The first is predicting which compounds microbes can metabolize, the second is connecting the chemistry of gut microbes to host phenotypes, and the third is linking gut chemistry to microbial ecology.

Towards the goal of systematically mapping the gut microbial chemistry that contributes to the metabolism of xenobiotics, including therapeutic drugs, recent efforts have used chemical structure-centric approaches to enable high-throughput computational predictions of gut microbe metabolism of drugs (*Sharma et al., 2017*; *Mallory et al., 2018*). These tools represent an important first step

towards ecological and mechanistic insights into gut microbiota driven biotransformation of foods and drugs. The second requirement, which has not yet been achieved, is to connect the known and predicted chemistry of gut microbes to host phenotypes. To date, information on human responses to therapeutic drugs is available in disparate databases and formats including FDA Adverse Report System (FAERs) (*Burkhart et al., 2015*), the Side Effect Resource (SIDER) (*Kuhn et al., 2016*) and DrugBank (*Law et al., 2014*). The third requirement, also lacking, is to systematically link gut microbe chemistry to microbial ecology to understand how the distribution of enzymes in populations of microbes facilitates ecological interactions that structure the human gut.

Here, we develop MicrobeFDT, a resource encompassing this 3-step framework that connects compound structure, enzyme function, taxonomy, and toxicity to characterize microbe-diet-drug-phenotype interactions. We organize ~10,000 food, drug, and endogenous compounds by structural similarity. We then link toxicity, enzyme interactions, and the propensity for gut microbes to carry out metabolism on each compound to the structural similarity network. We validate MicrobeFDT computationally by demonstrating that structural similarity is a reasonable proxy for toxicity, enzyme sharing, and coarse-grained functional similarity. We propose, and experimentally validate, active gut microbiome demethylation of an ovarian cancer drug, altretamine, a metabolism that we propose may drive toxicity of this drug. All data is available in the MicrobeFDT database (MicrobeFDT; *Guthrie, 2019*; copy archived at https://github.com/elifesciences-publications/microbeFDT-neo4j).

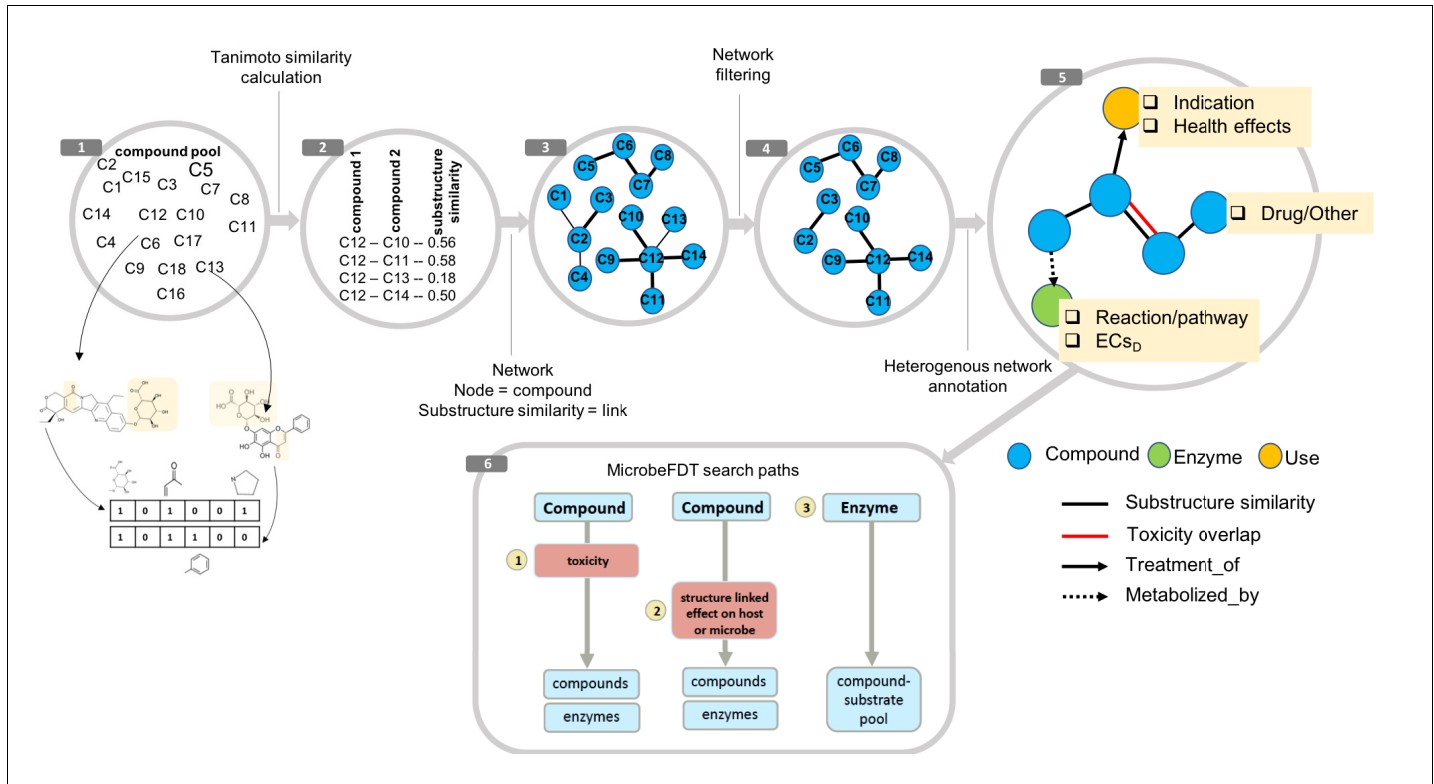

**Figure 1.** MicrobeFDT is a searchable resource of gut microbiome food and drug metabolism with associated toxicities. (1) Diet-derived, xenobiotic-derived and endogenous compounds were clustered based on the PubChem fingerprint system (*Kim et al., 2016*) and the Tanimoto coefficient (*Bajusz et al., 2015*). (2) The pairwise similarity matrix forms the basis of the (3) substructure similarity network in which nodes are compounds and links are weighted by substructure similarity. (4) A Z-score based threshold method was used to identify significant chemical similarity relationships between nodes (*Baldi and Nasr, 2010*). (5) The property graph model of nodes and relationships in the network highlights node-relationship pairs that can be queried. Node entities include compounds (blue), uses (orange) and enzymes (green). A compound node can have up to four types of directional relationships: compound pairwise substructure similarity, compound pairwise toxicity similarity, compound treatment use descriptor and compound microbial mediated metabolism descriptor.

DOI: https://doi.org/10.7554/eLife.42866.003

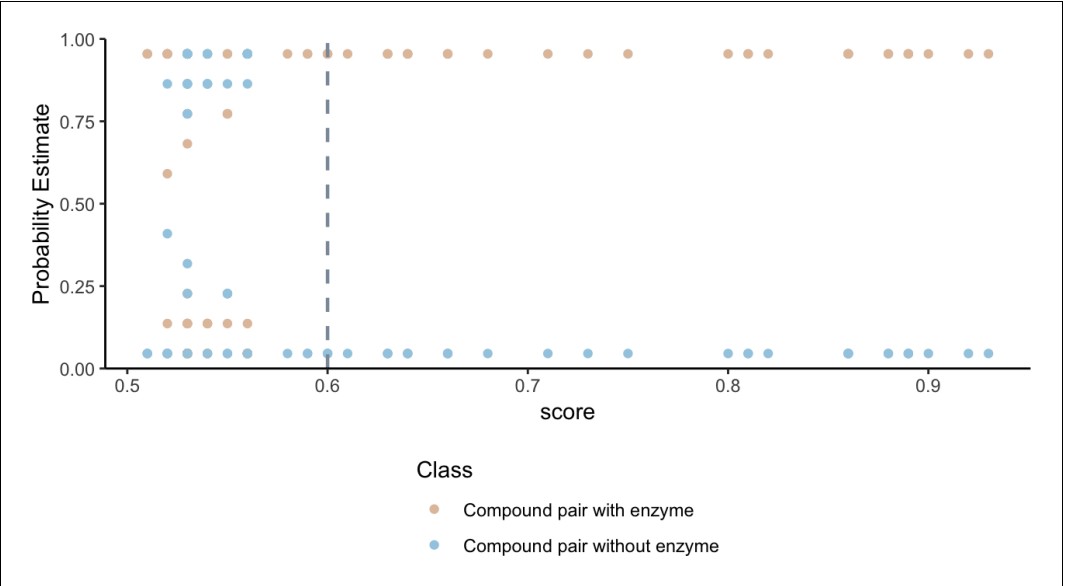

**Figure 2.** Higher substructure similarity scores between pairs of compounds are associated with higher probability of sharing an enzyme. Potential enzyme mediated metabolism of compound pairs is compared with substructure similarity to determine the probability that compounds have an experimentally determined shared enzyme (pink) or no known shared enzyme (blue). The gray vertical dashed line indicates the average cutoff for significance in substructure similarity neighborhood construction. Probability estimates are based on a Bayesian approach for support vector machines implemented in R using the probsvm package (*Zhang et al., 2013*).
DOI: https://doi.org/10.7554/eLife.42866.004

## Results

### Structural similarity as a metric to organize enzyme/taxonomy/toxicity links between compounds

The foundation of the MicrobeFDT resource is a chemical similarity network linking 10,822 food, drug, and endogenous compounds with PubChem compound identifier (CIDs) (*Kim et al., 2016*). In the network, nodes designate compounds and edges are weighted by pairwise chemical substructure similarity quantified by comparing PubChem fingerprints (*Kim et al., 2016*) using the Tanimoto score (*Bajusz et al., 2015*) (*Figure 1*). The Tanimoto score prioritizes overlap between compounds that share substructures over compounds with shared co-absences (*Bajusz et al., 2015*). We hypothesized that compounds with overlapping substructure and physiochemical properties, in which one compound is a known substrate of an enzyme, will be more likely to serve as substrates for the same enzyme. Recent *in silico* approaches to predict enzymatic reactions of drugs in the context of human enzyme catalyzed reactions also employ this hypothesis (*Niu et al., 2013*; *Yu et al., 2018*). Substructure-based clustering thus serves as a first step towards synthesizing publicly available information on gut compound chemical diversity and gut microbiome biochemistry.

To validate that our network can identify shared metabolism, we developed an in silico prediction model to assign a probability of shared metabolism between compounds based on substructure overlap and the following physiochemical categories: geometry, functional groups, amino acid composition, polarity and hydrophobicity. We find that the probability estimates of compound-pairs sharing an enzyme based on substructure and physiochemical parameters, increase as the substructure overlap score between compound pairs increases (*Figure 2*). Weighting compound pair chemical similarity relationships based on substructure similarity is thus a reasonable filtering step to identify compounds that may share metabolism.

As an example of how the network can reveal shared metabolism we selected compounds in the network with substructure overlap with digoxin, a cardiac glycoside inhibitor. Reduction of digoxin by a human microbiome reductase inactivates the drug, contributing to poor bioavailability in some individuals (*Haiser et al., 2013*; *Haiser et al., 2014*; *Lindenbaum et al., 1981*). Koppel et al.,

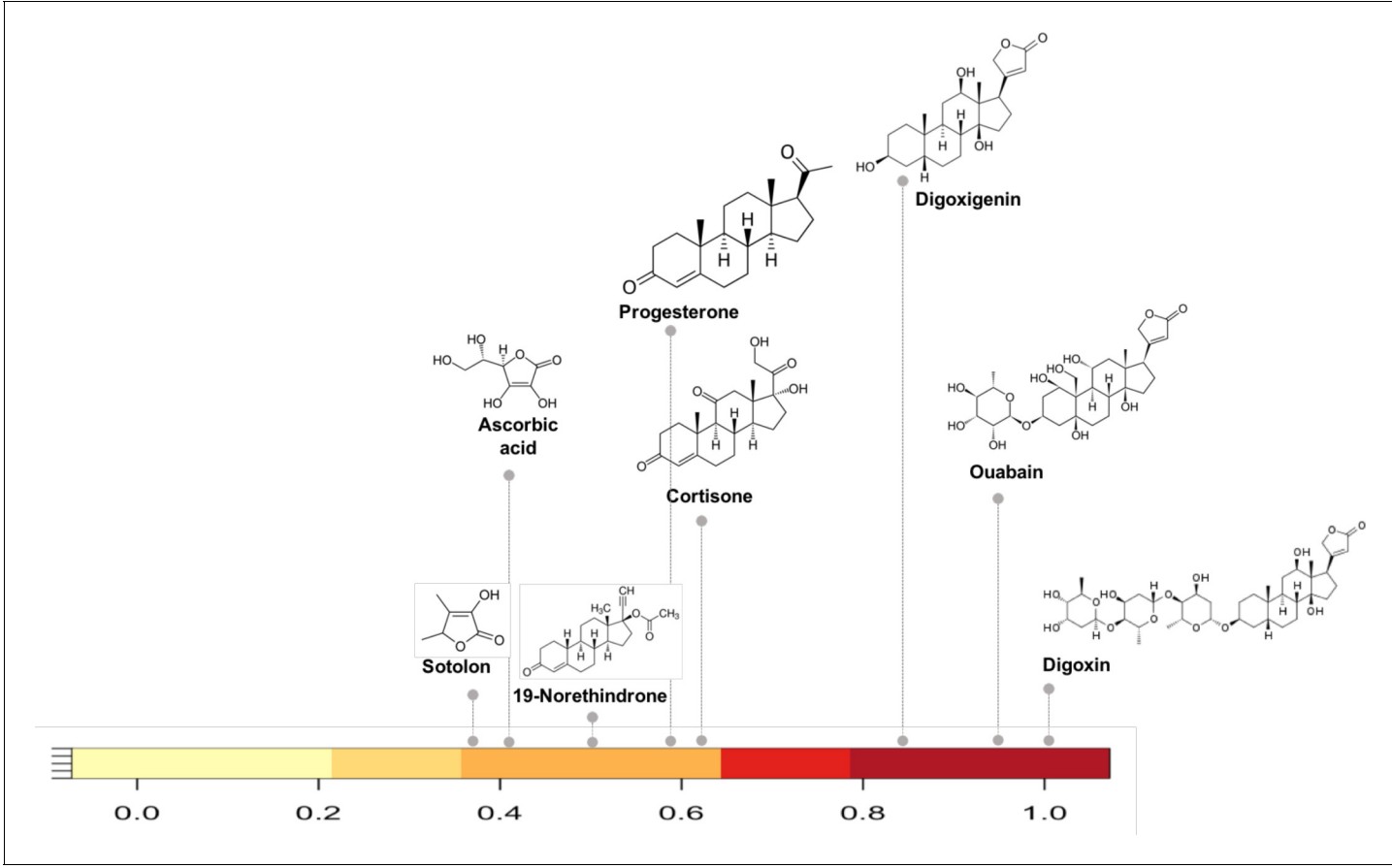

**Figure 3.** Substructure similarity range of Cgr2 enzyme susceptible compounds. Substructure based clustering distinguishes experimentally characterized substrates from non-substrates of the Cgr2 enzyme. Digoxin clusters with other cardenolides that are experimentally characterized substrates (*Koppel et al., 2018*) for Cgr2 at substructure similarity values greater than 0.8. Compounds that are not substrates of Cgr2 have lower substructure similarity with digoxin; compounds with minimal reduction (*Koppel et al., 2018*) include progesterone and cortisone (substructure similarity <= 0.63). Color bar intensity increases with compound overlap with digoxin.
DOI: https://doi.org/10.7554/eLife.42866.005

biochemically characterized the capacity of a single flavin- and [4Fe-4S] cluster-dependent reductase, *cgr2*, to reduce various substrates with a range of substructure similarity to digoxin (*Koppel et al., 2018*). We identified the substructure overlap between digoxin and compounds in the Koppel et al. study that were evaluated as substrates of Cgr2 enzyme. Among the biochemically assayed compounds (*Koppel et al., 2018*) that are present in the MicrobeFDT network, compounds with substructure similarity scores greater than 0.8 are also substrates for Cgr2. This assessment suggests that for the cgr enzyme substructure based clustering can distinguish experimentally characterized substrates from non-substrates (*Figure 3*).

Previous studies have found that structural similarity predicts both toxicity and drug target similarity (*Campillos et al., 2008*). To evaluate whether our network also recapitulates shared drug toxicity we fit a linear regression and computed the effect size to assess the association between substructure similarity and toxicity similarity for therapeutic drugs in our network. We find that structural similarity moderately positively predicts toxicity similarity for therapeutic drug pairs linked by structural similarity overall in the network (r = 0.03116, p<2.2e-16) (*Figure 4*).

Finally, we evaluated how well our compound clustering recapitulates structure-based chemical taxonomy as defined by the ClassyFire (*Djoumbou Feunang et al., 2016*) resource, a comprehensive chemical classification schema, at the level of superclass taxonomy. We found that substructure-based compound clustering, significantly groups compounds within a ClassyFire superclass based on a comparison of the MicrobeFDT network with a randomized network with the same number of

nodes and edges (p<8.06×10–15, Wilcoxon rank-sum test). Compound-pairs at higher substructure similarity share Superclass membership at higher substructure values and at a greater frequency than randomized pairs, indicating that the MicrobeFDT substructure similarity metric can capture established chemical classifications (*Figure 5*).

## Overlapping structural diversity of food, drug, and endogenous compounds

In the network, therapeutic drug structural diversity is embedded within food-derived chemical diversity. For example, drugs share structural similarity with food-derived compounds from a diverse range of classes including benzenoids, lipids, nucleosides and phenylpropanoids (*Figure 6*). Food derived compounds also contributed significantly greater molecular structure diversity (*Figure 6— figure supplement 1*) and higher self-similarity than therapeutic drug compounds (two-sample K-S test 0.49, p value=4.7395e-06).

## Assessing the distribution of enzymatic functions across taxonomic groups

Metabolic functions are not necessarily equally distributed across microbes in the microbiome. For example, as described above, inactivation of digoxin, a cardiac glycoside inhibitor, is linked to *cgr* operon expression levels in a single species, *E. lenta* (*Haiser et al., 2013*; *Koppel et al., 2018*). In contrast, the deconjugation and resulting reactivation of SN-38, the active metabolite of the chemo-therapeutic colorectal cancer drug irinotecan, is linked to a phylogenetically diverse guild of microbial β-glucuronidase carrying microbes (*Guthrie et al., 2017*; *Pollet et al., 2017*; *Wallace et al., 2015*).

The question arises, how many microbes can perform specific enzymatic functions? Knowing the taxonomic distribution of a function can guide approaches to validate hypotheses of microbiota driven modification of specific therapeutic drug or food compounds. More broadly, addressing this

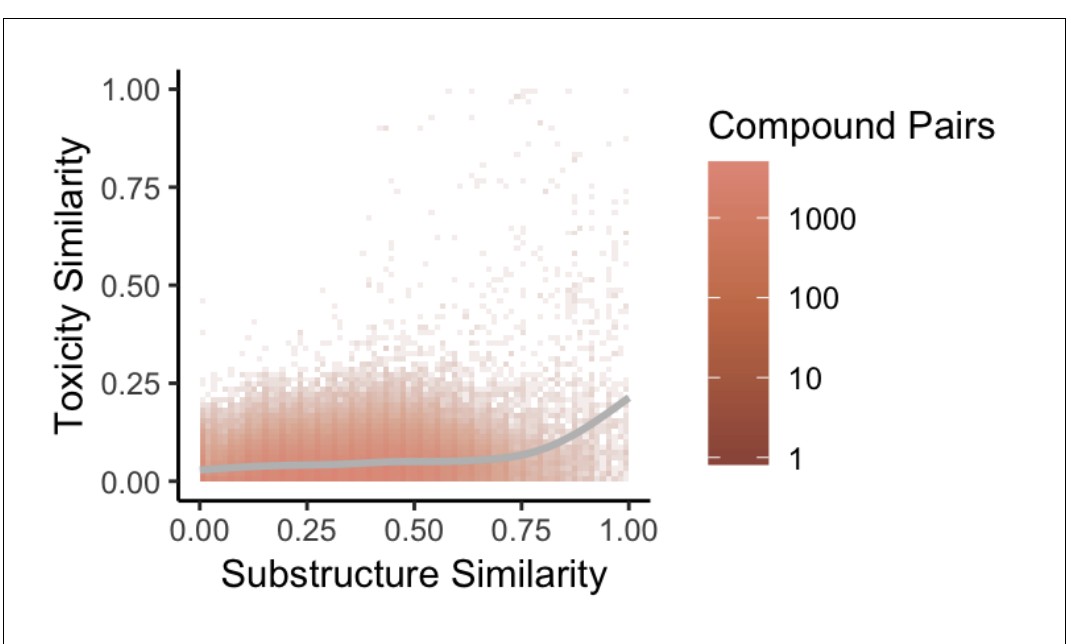

**Figure 4.** Substructure similarity is predictive of toxicity similarity. We evaluated the predictive power of substructure similarity to identify compounds with shared toxicity using a measure of pairwise toxicity defined by *Campillos et al. (2008)* and used a linear regression to determine the strength of the association. We find a modest positive correlation between substructure similarity and toxicity similarity that is stronger for more structurally similar compounds.

DOI: https://doi.org/10.7554/eLife.42866.006

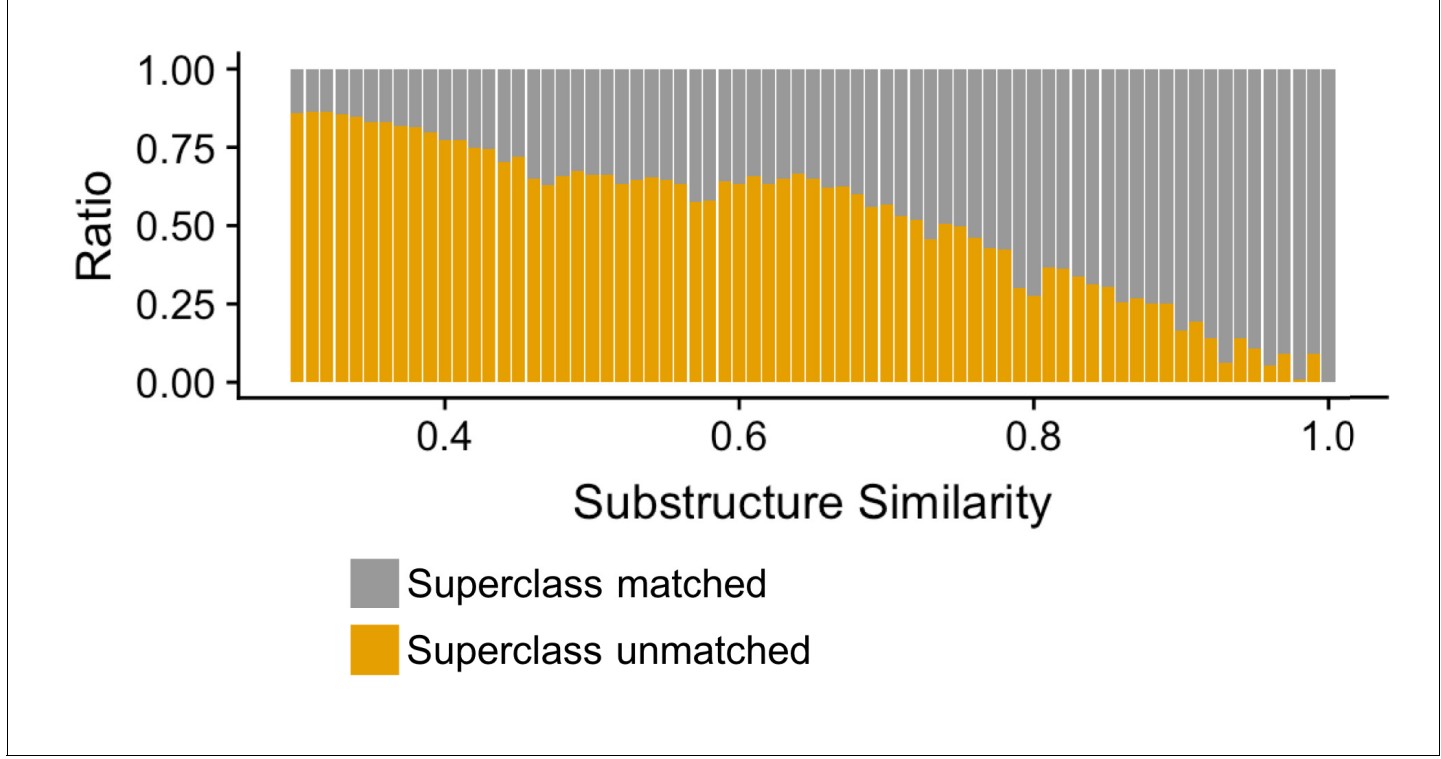

**Figure 5.** Compound-pairs share superclass annotation at a greater frequency as substructure similarity scores increase. Ratio of compound-pairs substructure similarity with matched and unmatched superclass annotation for all compound pairs represented in MicrobeFDT. Within the hierarchical ClassyFire classification schema, the superclass level annotation represents the second level and includes 31 different structure-based categories (*Djoumbou Feunang et al., 2016*).

DOI: https://doi.org/10.7554/eLife.42866.007

question informs therapeutic approaches for targeting specific enzymes to modulate patient responses to drugs and foods.

In MicrobeFDT, we quantify how many taxa have the capacity to carry out a specific function by applying a modified Simpson index function to compute an Enzyme Commission number-specific dominance ($ECs_D$) score for all enzymes present in the network. $ECs_D$ scores are based on the abundance of enzymes annotated at the species level across healthy human metagenomes from the Integrative Human Microbiome Project (iHMP) (*Proctor et al., 2014*) and are normalized between 0 and 1. Functions carried out by small numbers of species have values closer to 0 while functions carried out by taxonomically diverse groups have functions closer to 1. Thus, the $ECs_D$ indicates how broadly distributed a function is, a crucial metric for (1) understanding how to modify a function in the microbiome and (2) predicting how disruptive to the community modifying a function might be.

To validate $ECs_D$ scores we first identified biochemical pathways containing enzymes with high and low taxonomic dominance in the literature. Bacterial synthesis of various B group vitamins including biotin, cobalamin and riboflavin vary in the number of potential producers at the Phylum level (*Magnúsdóttir et al., 2015*). The most commonly synthesized B vitamin across diverse microbial taxa is riboflavin while vitamin B12 is dominated by Fusobacteria (*Magnúsdóttir et al., 2015*). The $ECs_D$ scores of cobalt-precorrin-2 C(20)-methyltransferase (0.305502) from the anaerobic Vitamin B12 synthesis pathway and riboflavin synthase (0.691618) from the riboflavin synthesis pathway in MicrobeFDT agree with the prior systematic genome assessment and experimental results of *Magnúsdóttir et al. (2015)* (*Figure 7*). While most bacteria do not synthesize sphingolipids, sphingolipid biosynthetic capacity has been identified in *Sphingomonas spp*, *Bacteroides* and human intestinal pathogens that synthesize and incorporate sphingolipids into their membranes or target host sphingolipids as a point of entry into host cell types (*Heaver et al., 2018*; *Heung et al., 2006*; *Olsen and Jantzen, 2001*). The low $ECs_D$ score of phosphatidate phosphatase (0.007353), an

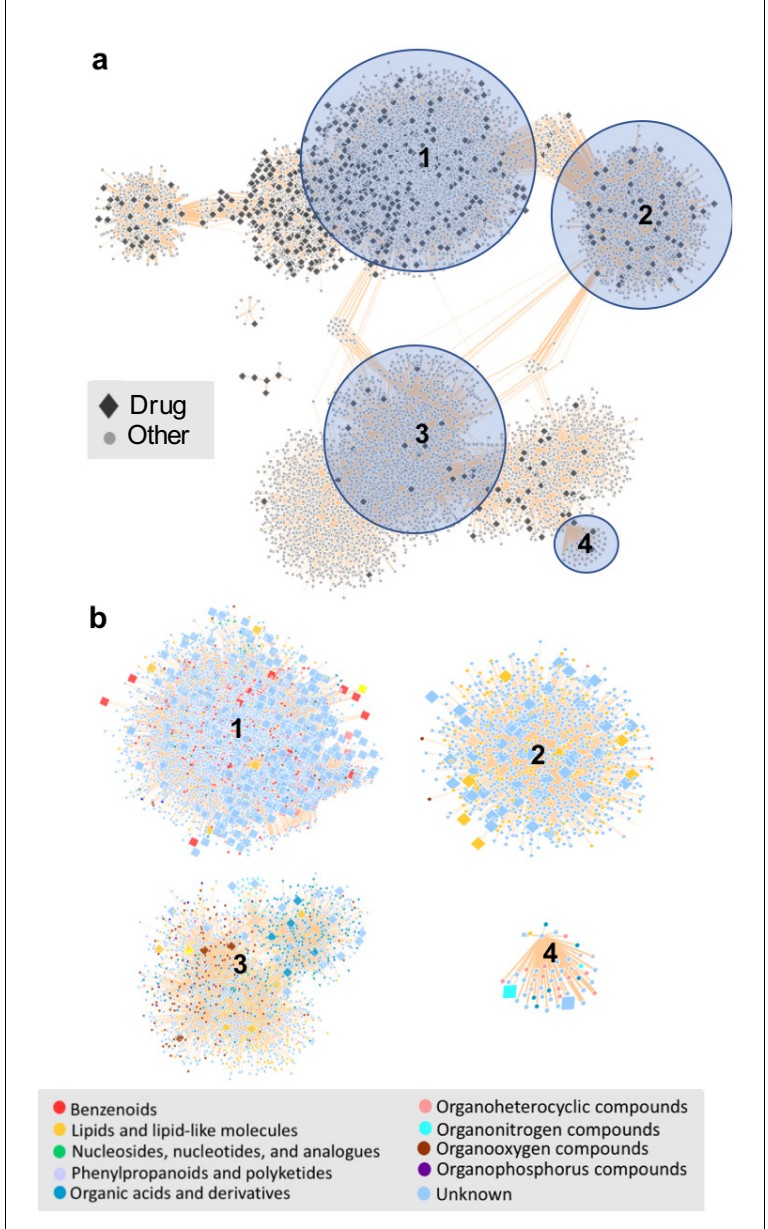

**Figure 6.** The chemical space of the gut microbiome. (a) Chemical similarity network of food-derived or endogenous compounds (gray circles, "Other") and therapeutic drugs (black diamonds, "Drug"). Tan edges are weighted by substructure similarity where thicker edges indicate higher substructure similarity. The distribution of compounds in chemical similarity space illuminates regions of low and high chemical substructure overlap between drugs and other compounds. (b) Compounds from selected regions of the network are colored by their superclass level taxonomy based on the FooDB chemical structure classification (*Wishart, 2012*). Food-derived or endogenously produced compounds are identified with blue circles, therapeutic drugs with red diamonds. Within high-drug density, highlighted regions **1** and **2,** drugs share substructure similarity with food-derived benzenoids, lipids, phenylpropanoids and polyketides. In the low-drug density highlighted region **3**, drugs overlap with organonitrogen compounds and nucleosides. Region **4** includes organonitrogen compounds and nucleosides in addition to lipid-like molecules which have minimal overlap with therapeutic drugs.
DOI: https://doi.org/10.7554/eLife.42866.008

The following source data and figure supplement are available for figure 6:

**Source data 1.** Chemical similarity scores for drug and non-drug compounds.
DOI: https://doi.org/10.7554/eLife.42866.010

*Figure 6 continued on next page*

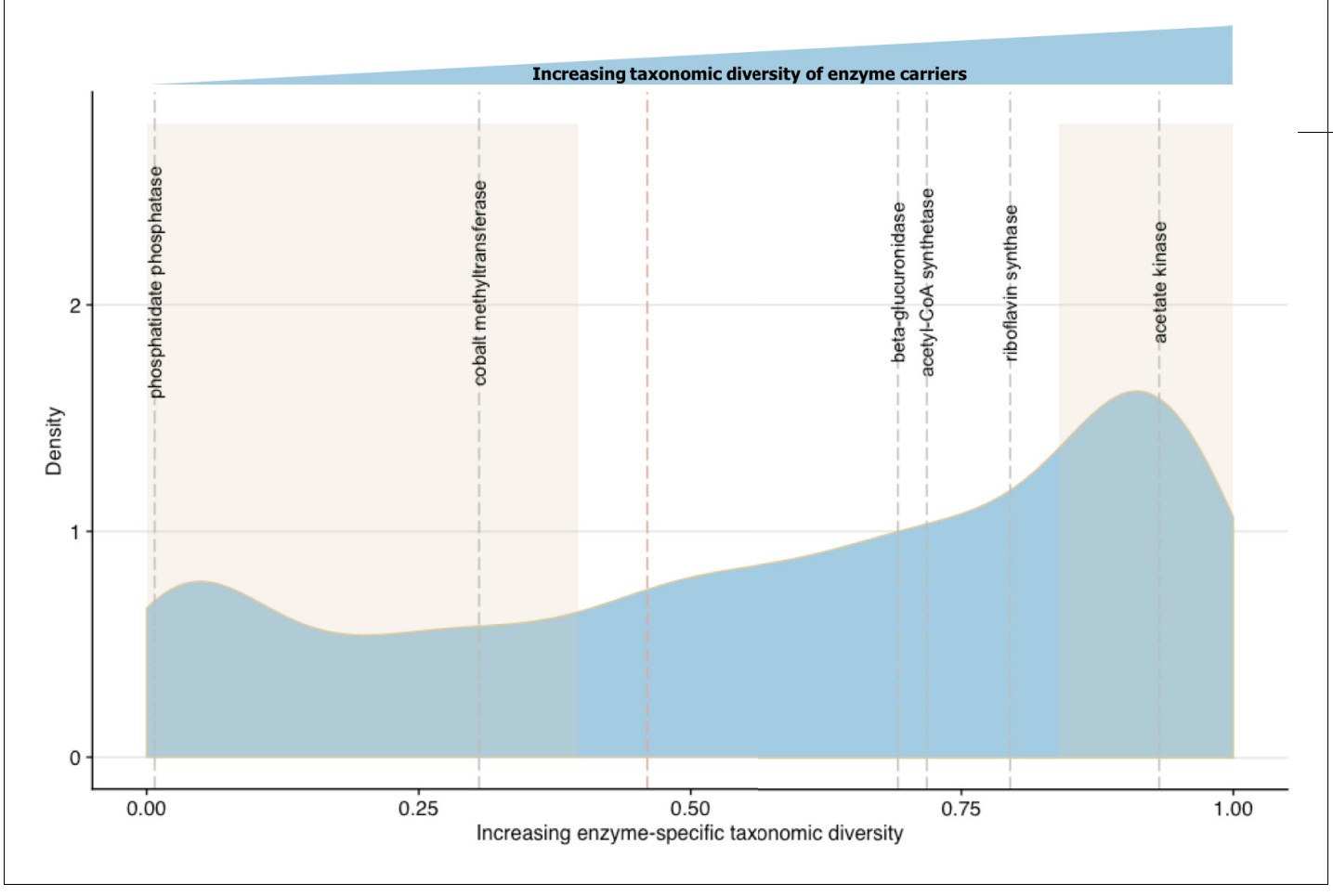

**Figure 7.** Linking enzymatic functions with taxonomic diversity. The Simpson index was adapted to describe enzyme-specific taxonomic dominance and diversity based on enzyme abundance in taxonomy-linked gene counts across healthy individuals in the Integrative Human Microbiome Project (*Proctor et al., 2014*). We define a microbial enzyme as high dominance and low taxonomic diversity if its Simpson index value falls below 0.46 (red dotted line), the mean value across all enzymes. Dominance-diversity values for gut microbiota functions that fall above or below the mean are highlighted by gray dashed lines and include the following enzymes and pathways: phosphatidate phosphatase (0.007353), cobalt-precorrin-2 C(20)-methyltransferase (0.305502) from the Vitamin B12 synthesis pathway, β-glucuronidase (0.691618), Acetyl-CoA synthase (0.718163) which is involved in the production of propionate from complex carbohydrates, riboflavin synthase (0.794781) from the riboflavin synthesis pathway and acetate kinase (0.931892) which is involved in acetate production. The shaded regions indicate the range of $EDs_D$ values that are one standard deviation above and below the mean and reflect the most broadly distributed functions and most specialized functions.

DOI: https://doi.org/10.7554/eLife.42866.011

enzyme involved in sphingolipid biosynthesis and metabolism (*Olsen and Jantzen, 2001*), mirrors the limited distribution of the sphingolipid biosynthetic capacity across gut microbes.

## Combining chemical and toxicity similarity to predict microbial N-demethylase contribution to drug metabolism and toxicity

To provide a practical example of using multiple features of MicrobeFDT to identify uninvestigated microbiota-driven drug toxicity, we searched the network for compounds with high structural and toxicity similarity. Among these compounds were the ovarian cancer drug altretamine (*Lee and Faulds, 1995*) and the environmental contaminant melamine (*Figure 8*). Both melamine and altretamine have toxicity profiles that include diarrhea and renal toxicity (*Rose et al., 1996*; *Zheng et al., 2013*). Melamine, an industrial compound, has experimentally validated microbiome-mediated toxicity (*Zheng et al., 2013*). Altretamine toxicity, however, has not previously been linked to an

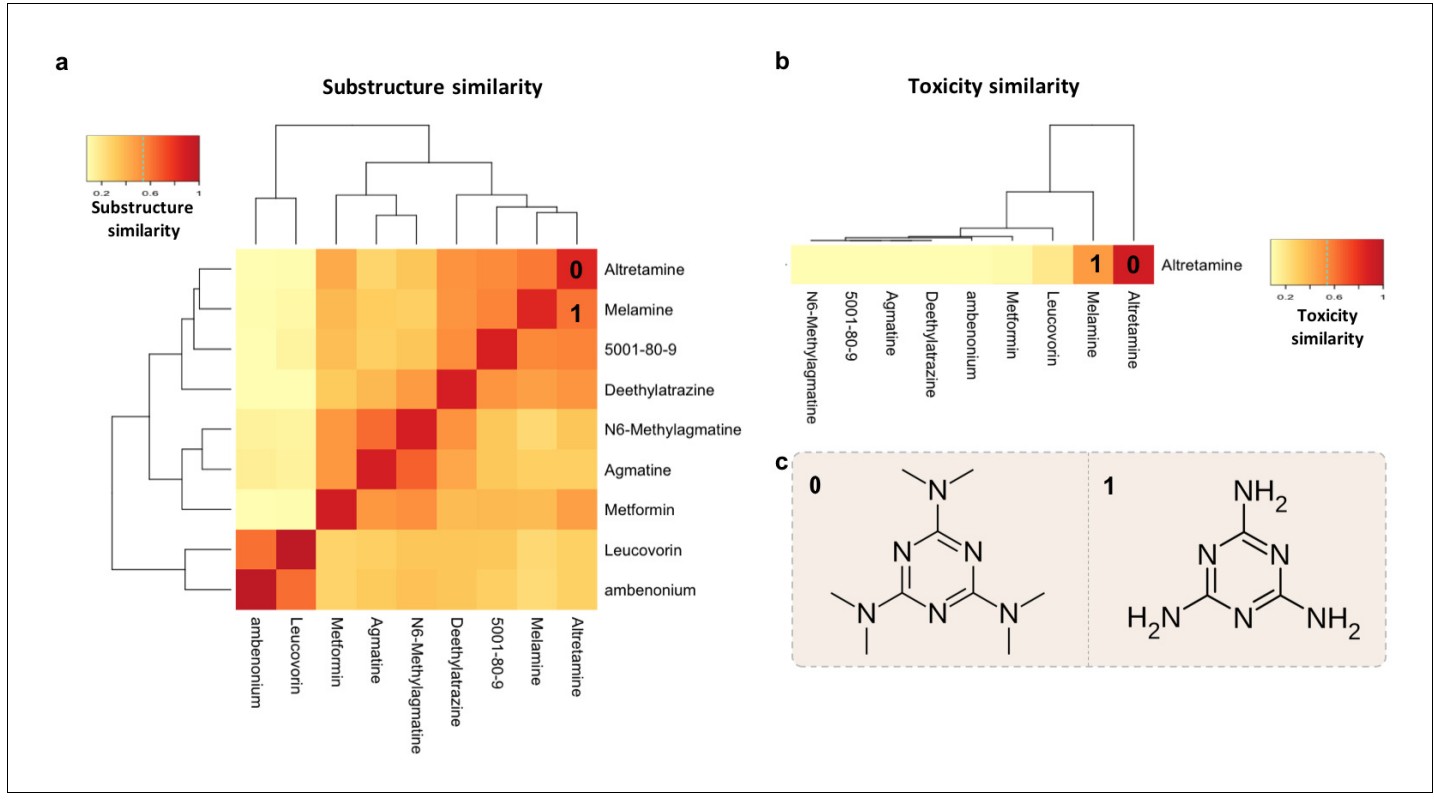

**Figure 8.** Structure-toxicity relationship between melamine and altretamine suggests a role for microbial N-demethylases in altretamine toxicity. (**a**) Substructure overlap between altretamine and its nearest neighbors in MicrobeFDT. A Z-score based threshold of significant overlap indicates that altretamine has both high substructure and (**b**) toxicity overlap with melamine. (**c**) The two compounds are distinguishable by the presence of N-methyl groups.

DOI: https://doi.org/10.7554/eLife.42866.012

The following figure supplement is available for figure 8:

**Figure supplement 1.** Phylogenetic distribution of N-demethylases in healthy human guts.
DOI: https://doi.org/10.7554/eLife.42866.013

individual's gut microbiota. Approximately half of patients taking altretamine orally experience various forms of gastrointestinal toxicity including diarrhea, nausea and/or vomiting (*Keldsen et al., 2003*).

Within the network altretamine is linked to microbial N-demethylase enzymes which may remove methyl groups from this compound, potentially leading to similar toxic effects as seen with melamine. We found no published experimental evidence of gut microbiota mediated conversion of altretamine. However, N-demethylases in *Pseudomonas putida CBB5* enable this microbe to grow on caffeine and other purine alkaloids as the sole carbon and nitrogen source; thus annotated N-demethylases in *P. putida CBB5* can act on compounds that are structurally similar to altretamine (*Summers et al., 2012*). Furthermore, we identify hypothetical proteins homologous to *Pseudomonas putida CBB5* N-demethylases in a subset of healthy human guts (*Figure 9—figure supplement 1*). We hypothesized that gut microbial N-demethylases may partially or completely N-demethylate altretamine, converting it into metabolites that contribute to patient toxicity.

A first step in validating this hypothesis is to demonstrate that the gut microbiome can demethylate altretamine. We incubated altretamine in a pooled fecal slurry generated from three healthy individuals and monitored altretamine and potential metabolites using LC-MS. We controlled for the formation of spontaneous N-demethylation of altretamine, which has been reported in the literature (*Damia and D'Incalci, 1995*), and found that a metabolite that is structurally identical to pentamethylmelamine, a demethylated altretamine metabolite, increases in active fecal microcosms over 48 hr (*Figure 9*). In active fecal biotic conditions the metabolite continually increased between

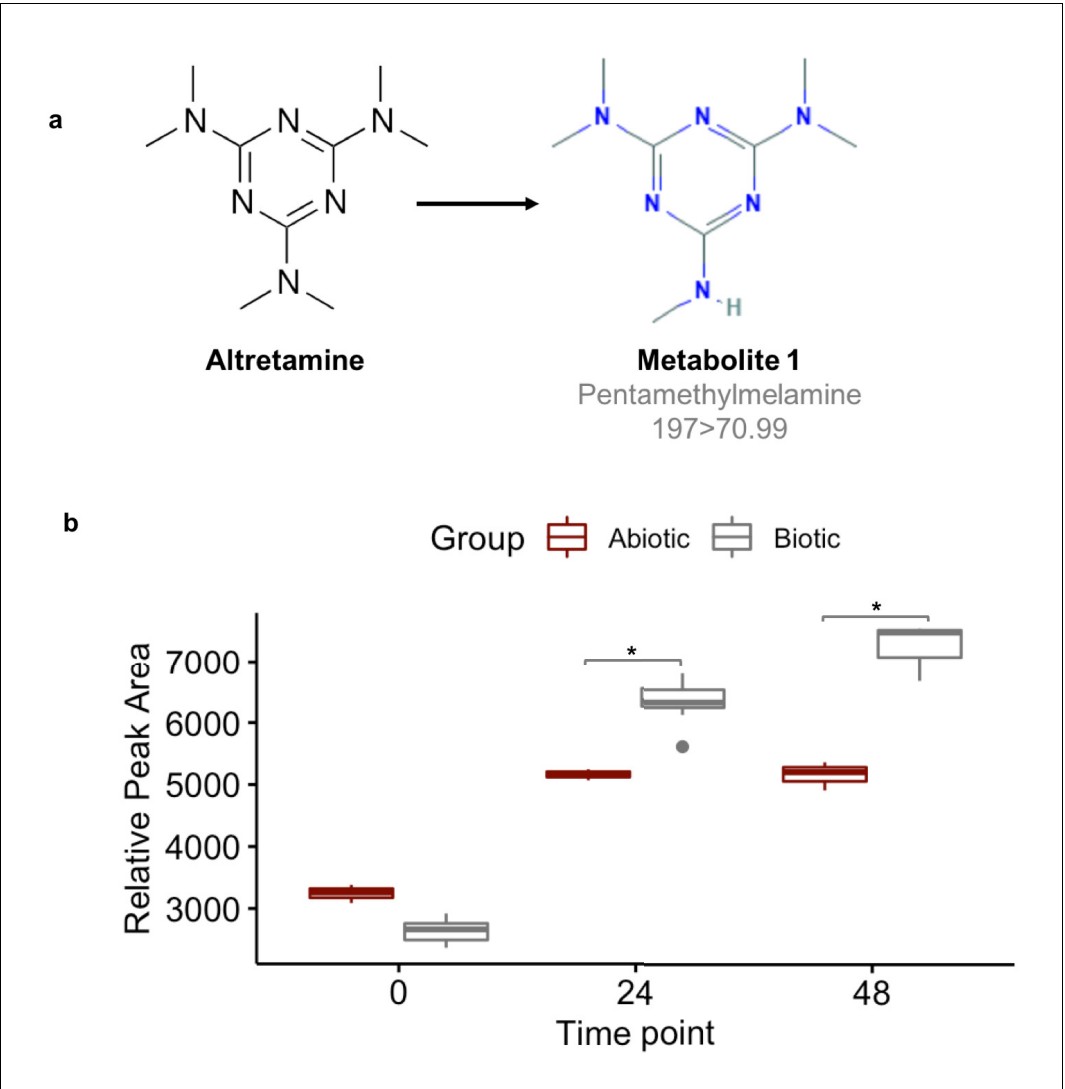

**Figure 9.** Fecal microbiomes actively demethylate altretamine. Liquid chromatography with tandem mass spectrometry (LC-MS) was used to quantify the formation of (**a**) pentamethylmelamine, an N-demethylated metabolite of altretamine identified in the pooled fecal microbiomes of three healthy unrelated individuals. (**b**) The formation of metabolite 1 at 24 and 48 hr was significantly increased under the experimental condition in comparison to the contribution of spontaneous N-demethylation by an unpaired two-sample Wilcoxon test (*=$P < 0.05$).

DOI: https://doi.org/10.7554/eLife.42866.014

The following figure supplement is available for figure 9:

**Figure supplement 1.** Experimental design and controls used to quantify fecal microbiome turnover of altretamine.

DOI: https://doi.org/10.7554/eLife.42866.015

time 0 and 48 hr. Killed controls demonstrated an increase in metabolite between 0 and 24 hr, though to a lesser extent than in active fecal microcosms. Notably there was little metabolite formation after 24 hr, indicating that in addition to abiotic N-demethylation, active gut microbes demethylate altretamine to the putative metabolite pentamethylmelamine.

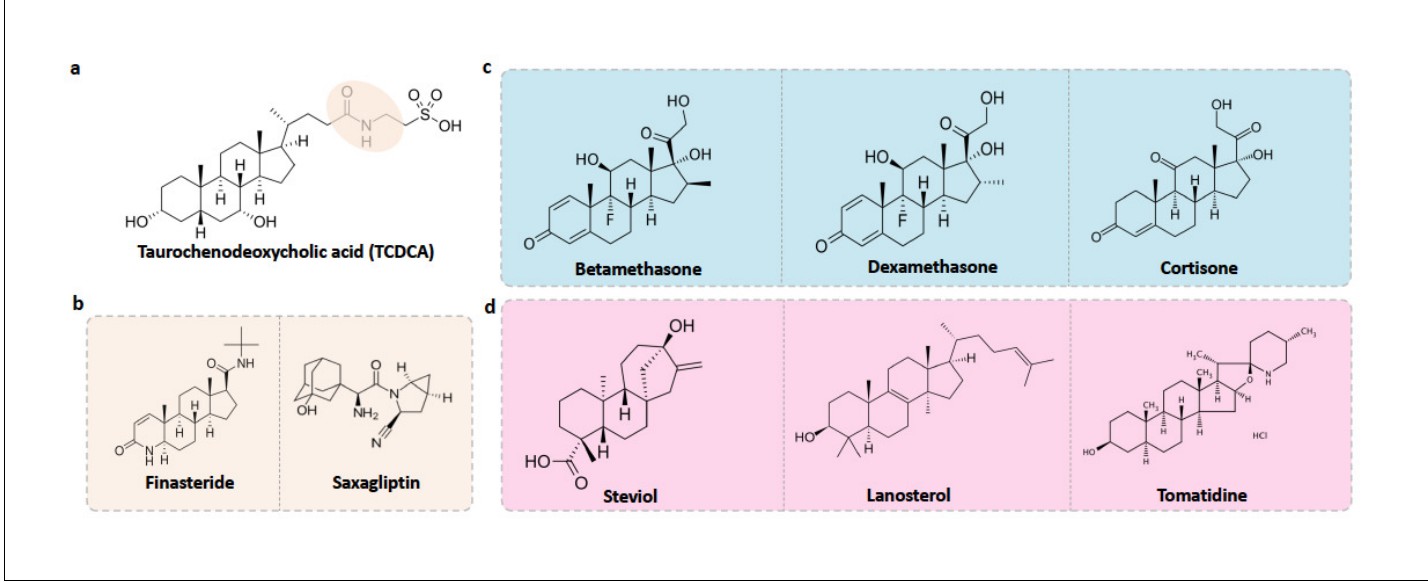

**Figure 10.** Food-drug compounds chemically similar to TCDCA are putative antimicrobials. (a) Chemical structure of taurochenodeoxycholic acid (TCDCA). (b) TCDCA-like therapeutic drugs that are susceptible to bile salt hydrolases include finasteride and saxagliptin. (c) Non-susceptible TCDCA-like therapeutic drugs include betamethasone, dexamethasone and cortisone. (d) TCDCA-like food derived compounds include steviol, lanosterol and tomatidine.

DOI: https://doi.org/10.7554/eLife.42866.016

The following figure supplement is available for figure 10:

**Figure supplement 1.** Phylogenetic distribution of bile salt hydrolases in healthy human guts.
DOI: https://doi.org/10.7554/eLife.42866.017

## Food derived compounds and non-antibiotic therapeutic drugs with potential antimicrobial properties

MicrobeFDT suggests an unrecognized role for bile acid-like foods and drugs in altering the composition of the human gut. Conjugated primary bile acids (BA) function as potent detergents and antimicrobial agents capable of dissolving microbial membranes and causing intracellular acidification; bile acid function is linked to specific structural features of these compounds (*Jones et al., 2008*; *Begley et al., 2006*). Taurochenodeoxycholic acid (TCDCA) is a taurine conjugated primary bile acid with a diet-tunable concentration in the gut (*Ridlon et al., 2016*). Energy drinks, animal protein and fish are rich sources of taurine while vegetarian and vegan diets dominated by fruits, vegetables, legumes and soy are poor sources (*Ridlon et al., 2016*). Taurine conjugated bile acids are hypothesized to contribute to the etiology of colorectal cancer by generating hydrogen sulfide during microbial mediated de-conjugation of taurine conjugates (*Ridlon et al., 2016*). Conjugated primary bile acids have demonstrated in vitro activity as antimicrobial compounds, for example glycocholic and taurocholic conjugated bile acids are bacteriostatic, inhibiting *S. aureus* growth by decreasing intracellular pH and disrupting the proton motor force (*Sannasiddappa et al., 2017*).

Using MicrobeFDT, we identified therapeutic drug and food compounds that are structurally similar to TCDCA; we propose these compounds might have similar antimicrobial effects on the microbiome and we discuss studies from other groups that support this hypothesis (*Figure 10a*).

Bile salt hydrolase (BSH) mediated bile salt deconjugation is one mechanism that gut microbes use to detoxify conjugated primary bile acids (*Begley et al., 2006*); thus BSH activity may support gut bacterial persistence in face of frequent contact with primary BAs. We first subdivided TCDCA-like antimicrobial compounds based on BSH enzyme susceptibility. BSH enzymes are phylogenetically diverse and abundant across healthy human fecal metagenomes (*Figure 10—figure supplement 1*). Among the BSH-susceptible therapeutic drug compounds, we identified known antibiotics such as clindamycin and lincomycin, as well as non-antibiotic prescribed therapeutics such as finasteride, which is used for the treatment of androgenetic alopecia (*Manabe et al., 2018*) and benign prostatic hyperplasia (*Chau et al., 2015*), and the oral antidiabetic drug saxagliptin (*Men et al.,*

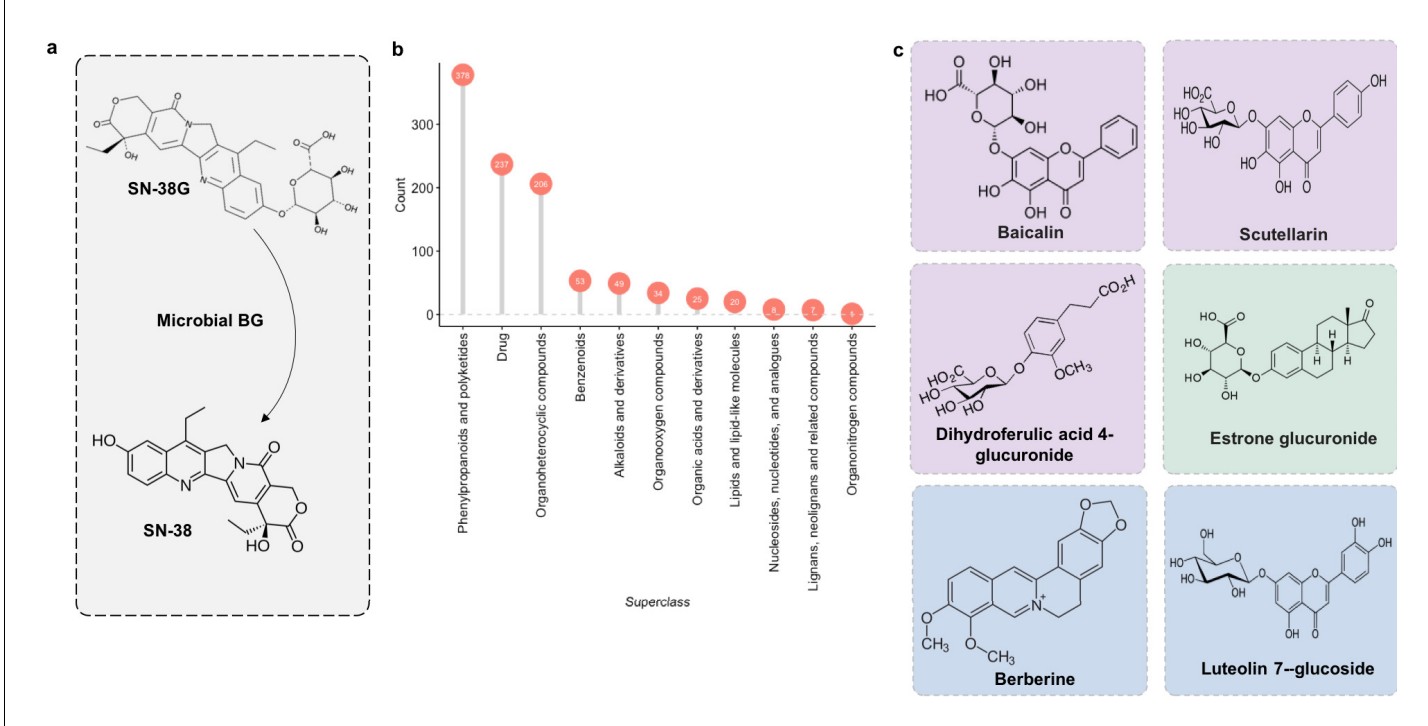

**Figure 11.** Microbial β-glucuronidase potential substrate pool of compounds structurally similar to SN-38G. (**a**) SN-38G conversion to SN-38 in the gut is mediated by microbial β-glucuronidases. (**b**) The substrate pool for β-glucuronidases with above threshold substructure overlap with SN-38G are members of a diverse range of chemical structure superclasses as defined by FooDB chemical ontology (**Wishart, 2018**). (**c**) These compounds include glucuronidated food-derived compounds (purple), endogenous glucuronides (tan) and other non-glucuronides (blue).
DOI: https://doi.org/10.7554/eLife.42866.018

**2018**) (**Figure 10b**). Notably, in a Wistar rat model of chronic bacterial prostatitis (CBP), finasteride reduces bacterial infection as a single agent and has a synergistic effect with ciprofloxacin through an unknown mechanism (**Lee et al., 2011**). Through in vitro studies, Chavex-Dozal and colleagues propose a role for finasteride in the prevention of *Candida albicans* biofilm formation and filamentation (**Chavez-Dozal et al., 2014**). These experimental results support the hypothesis that finasteride may have unrecognized off-target antibiotic effects.

Most TCDCA-like compounds in MicrobeFDT are non-BSH susceptible food-derived compounds. Among the TCDCA-like non-BSH susceptible compounds are oral steroid medications, including dexamethasone and betamethasone (**Figure 10c**). The immunomodulatory activities of glucocorticoids, including dexamethasone, involve the activation of genes related to anti-inflammatory cytokines such as IL-10 and proteins that inhibit the pro-inflammatory NFκB signaling pathway (**Coutinho and Chapman, 2011**; **Huang et al., 2015**). Dexamethasone has known anti-microbial properties. For example, dexamethasone has dose-dependent anti-microbial activity against clinically isolated *Streptococcus milleri*, *Aspergillus flavus*, and *Aspergillus fumigatus* in culture, while not killing *Staphylococcus aureus* (**Neher et al., 2008**). *Pseudomonas aeruginosa* was found to be susceptible to dexamethasone at high concentrations (**Neher et al., 2008**). Cortisone, which also has significant structural overlap with TCDCA, has been linked to a variety of opportunistic infections by enteric bacterial pathogens, for example an increase in gastrointestinal parasites (**Nair et al., 1981**) and reactivation of *Chlamydia pneumoniae* (**Laitinen et al., 1996**).

Food-derived TCDCA-like compounds include steviol, lanosterol and tomatidine. Steviol is a component of stevia which has antimicrobial properties against *Borrelia burgdorferi in vitro* (**Theophilus et al., 2015**), and lanosterol derivatives have antifungal activities (**Shingate, 2013**). Tomatidine was recently identified as an antibiotic molecule that inhibits ATP synthesis against *Staphylococcus aureus* (**Lamontagne Boulet et al., 2018**); we hypothesize that the antimicrobial activity of this compound may include intracellular acidification given its structural overlap with

TCDCA (*Figure 10d*). The network thus identifies compounds with known anti-microbial properties in addition to proposing additional, structurally related compounds with uncharacterized effects. We propose that in addition to modulating immune responses, bile salt-like compounds may selectively alter human microbiomes, again, with unknown consequences for treatment outcomes and health.

### MicrobeFDT identifies the diet-derived substrate pool for microbial BGs and candidates for nutritional competition with SN-38G

We next applied MicrobeFDT to identify diet-derived substrates of a gut carbohydrate active enzyme, β-glucuronidase. β-glucuronidases play a major role in the toxicity of the colorectal cancer chemotherapeutic prodrug irinotecan (CPT-11), whose active form, SN-38, is inactivated by hepatic glucuronidation and excreted into the gut as the inactive metabolite SN-38 glucuronide (SN-38G) (*Wallace et al., 2010*; *Sparreboom et al., 1998*). Microbial β-glucuronidases hydrolyze the glucuronide group, releasing the aglycone SN-38 into the intestinal environment (*Figure 11a*). Deconjugation promotes epithelial damage and severe diarrhea in some patients and in mouse models (*Wallace et al., 2010*; *Sparreboom et al., 1998*; *Slatter et al., 2000*).

We previously demonstrated that individual human fecal samples have variable capacities to deconjugate SN-38G (*Guthrie et al., 2017*). Identifying the full substrate pool of β-glucuronidases is thus important for 1) understanding how diet contributes to β-glucuronidase abundance and expression levels in the gut and 2) to enable novel therapeutic strategies such as nutritional competition.

Some food compounds may be preferred substrates for microbiome β-glucuronidases which would otherwise deconjugate SN-38G. If true, one could potentially alleviate toxicity associated with the deconjugation of SN-38G via nutritional competition with a preferred substrate. Therefore, we scanned the chemical similarity module containing SN-38G for dietary compounds that may serve as alternative substrates for microbial β-glucuronidases. Most compounds identified as significantly similar to SN-38G were food derivatives or other constituents (*Figure 11b*). Among these targets were flavonoids such as baicalin and scutellarin which are widely distributed in plants (*Kumar and Pandey, 2013*) (*Figure 11c*). We propose that these compounds may compete with SN-38G for turnover by microbial β-glucuronidases and are a potential avenue for decreasing the adverse drug responses associated with irinotecan administration.

## Discussion

The chemical space of the human gastrointestinal tract ecosystem is shaped by host dietary intake, xenobiotic exposure, and host and gut microbiome derived products. In turn, diet shapes the composition and potential niches of organisms within human gut microbiomes. A combination of compound, host, and microbiome features influence potential microbial metabolism. Examining these features individually cannot reliably infer clinical phenotypes associated with microbiome/compound interactions. Two molecules may have the same toxicity profile but very different biochemistry, for example. Automated enzyme annotation may be incorrect, and compound structural similarity is often insufficient to predict substrate preferences. Finally, enzymes that carry out a reaction associated with a patient phenotype may be unevenly distributed across microbes and across human microbiomes. MicrobeFDT is designed to overcome some of these limitations by enabling a more holistic analysis of toxicity, structure, metabolism and ecology. We used a combination of network features to successfully predict the novel microbial metabolism of the cancer drug altretamine.

Metabolomics data indicate active demethylation of altretamine by fecal slurries but cannot propose a mechanism by which microbial activity metabolizes this compound. MicrobeFDT suggests that altretamine is a putative substrate of microbial N-demethylases. Microbe-mediated N-demethylation reactions, and the subsequent release of N-methyl groups, occur as a part of amino acid and nucleotide metabolism (*D'Mello and International, 2017*). Notably, diet is a source of amino acids which are derived in part from metabolism of dietary choline, carnitine and legumes, and have physiological functions for bacteria including osmoprotection and incorporation into bacterial flagellin proteins and lipid membranes (*Goldfine and Hagen, 1968*). Amino acid-specific bacterial N-demethylases have been identified but are poorly characterized (*Wargo, 2017*). Additionally, fecal and species specific N-demethylation has been observed for other therapeutic drugs and commonly ingested compounds such as caffeine, which clusters with altretamine in the network due to its structural similarity (*Summers et al., 2012*; *Caldwell and Hawksworth, 1973*; *Clark et al., 1983*;

*Colombo et al., 1982*). N-demethylases can act on chemically diverse substrates (*Wargo, 2017*; *Burnet et al., 2000*). Given this body of evidence, we propose N-demethylases may demethylate altretamine partially or completely, creating metabolites that are toxic to patients.

Human gut metagenomic data indicate that Rieske family oxidative N-demethylases are carried by a small, phylogenetically conserved set of gut taxa, with notable inter-personal variation. That these enzymes require oxygen may make them more relevant during disruptions to gut homeostasis when oxygen becomes available, such as colonic crypt hyperplasia caused by injuries to the intestinal epithelia (*Litvak et al., 2018*). Finally, we note that N-demethylation in the gut may be relevant for differences in individual metabolism of numerous other compounds such as the cancer drug tamoxifen, the widely used antihistamine diphenhydramine, and theobromine, a plant alkaloid found in foods. While is possible that N-demethylation is enzyme independent or that enzymes annotated with other functions are responsible for this activity, MicrobeFDT provides a clear path forward for mechanistic studies of N-demethylation in the gut.

Beyond predicting the toxicity or function of gut compounds, MicrobeFDT identifies the larger substrate pool for enzymes involved in drug metabolism. For example, shared conjugation patterns may represent a clinically relevant way to group compounds that share microbial enzymatic processing. As an example, compounds inactivated by glucuronidation are susceptible to microbial β-glucuronidase-mediated reactivation. We used MicrobeFDT to identify compounds structurally similar to the conjugated, detoxified irinotecan metabolite SN-38G and found dietary substrates that may interact with similar β-glucuronidases that this drug interacts with. Structurally similar compounds may act competitively – via inhibition of SN-38G turnover by higher priority β-glucuronidase substrates or synergistically – via substrate inducible transcriptional upregulation of β-glucuronidase enzymes. A person consuming a large amount of the plant-based compound scutellarin as part of a supplement, for example, might be inadvertently modulating the effects of their cancer therapy.

Outside of drug metabolism, β-glucuronidases mediate deconjugation and enterohepatic circulation of estrogens, impacting the human host total estrogen burden (*Shapira et al., 2013*; *Kwa et al., 2016*). It has been hypothesized that β-glucuronidase deconjugation may result in greater absorption of estrogens and thus influence the development of estrogen-driven cancers including breast, ovarian and endometrial cancers (*Shapira et al., 2013*; *Kwa et al., 2016*). Our network is useful for developing mechanistic hypotheses targeting how diet and the microbiome jointly act as moderators of estrogen-driven cancers, and to suggest opportunities for diet-based modulation of total estrogen levels.

An important step towards characterizing the role of the gut microbiome in shaping individual responses to foods and drugs is identifying how gut microbiome metabolism varies from compound to compound and how this metabolism relates to inter-personal variation in diet or drug responses to specific compounds. To tackle this challenge, we add the context of taxonomic diversity to the predicted impact of microbial on specific targets by quantifying enzyme specific taxonomic dominance and diversity with a novel metric, the $ECs_D$ score. This score distinguishes enzymatic activities carried out by single species or few taxa, such as N-demethylase activity, from those where many taxa may contribute, such as β-glucuronidase and bile salt hydrolase activity. The $ECs_D$ score is a readout of potential substrate metabolism at the community level that can be linked to inter-personal variation in gut function and phenotypic outcomes.

The structural similarity network that underlies MicrobeFDT could be improved by using compound atom and bond connectivity information as an additional filtering step for compounds of interest, for example by using information from the SMARTS molecular pattern matching language (*Chepelev et al., 2012*). SMARTS can be used to specify sub-structural patterns in molecules; these patterns could be added to MicrobeFDT as an additional information source indicating potential active moieties in compounds.

MicrobeFDT does not predict substrate specificity for microbiome enzymes; available data and methods are not sufficient to achieve this goal. Enzyme promiscuity also shapes the probability that two chemically overlapping compounds will be processed by the same enzyme. A future improvement to our resource could extract data from resources like RetroRules (*Duigou et al., 2019*), which uses SMARTS strings to define reaction rules, or utilize the Promis server measure of enzyme multifunctionality (*Carbonell and Faulon, 2010*) to further support a user's ranking of hypothesized compound-enzyme interactions.

It must be noted that the set of diet-derived and xenobiotic compounds that form the basis of the network is a non-exhaustive representation of the gut chemical landscape. Efforts to characterize the gut chemical space using metabolomics approaches including mass spectrometry and nuclear magnetic resonance spectroscopy will play key roles in elucidating a fuller gut chemical landscape (*Vernocchi et al., 2016*; *Wishart, 2012*). MicrobeFDT does not address the issue of compound concentrations in the gut, which are vital to assess likely physiological effects. Lastly, MicrobeFDT is limited to enzymes in KEGG, and does not address the many hypothetical enzyme sequences identified through metagenomic sequencing. Despite these limitations, MicrobeFDT highlights areas of known gut chemical space for which our understanding of microbial processing is limited and is a powerful tool to guide mechanistic investigations into diet-drug-microbiota interactions.

# Materials and methods

**Key resources table**

| Reagent type (species) or resource | Designation | Source or reference | Identifiers | Additional information |
|---|---|---|---|---|
| Biological sample (community microbiota, feces) | fecal sample | other | | fecal sample obtained from three healthy adults |
| Chemical compound, drug (altretamine) | altretamine | Sigma | Pubchem_ID:329748966; CAS_No:645-05-6 | prepared in DMSO, 0.1 mM final concentration in fecal slurry |
| Other | Brain Heart Infusion broth | Himedia | Himedia:M210I | |
| Chemical compound (Dimethyl sulfodixe) | DMSO | MP Biomedicals | MP:191418; CAS_No:67-68-5 | |
| Chemical compound (Melamine-triamine-(15N3)) | Melamine-triamine-($^{15}N_3$) | Sigma | Pubchem:329758619; CAS_No:287476-11-3 | prepared in DMSO, 400 nM final concentration in analytical sample |

## MicrobeFDT pipeline

The MicrobeFDT graph database encodes heterogeneous information on the interactions between compounds and microbial enzymes in the gut chemical landscape, highlighting the following four relationships across 13,440 nodes (10,822 xenobiotic, diet-derived and human gut endogenous compounds, 2062 microbial enzymes and 525 therapeutic drug use labels) defined from publicly available data directly or computed: (1) compound-compound substructure similarity; (2) compound-compound toxicity similarity; (3) microbial enzyme-compound interactions; and (4) drug-indication associations. The database is implemented in Neo4j (https://neo4j.com/) and can be queried through the Cypher Query Language. Through graph-based searches users can query the network based on node or relationship features. MicrobeFDT can be accessed here (*Guthrie, 2019*).

## Publicly available datasets and resources used as inputs for the network

The SIDER 4.1 side effect resource is a database of approved medicines and their known adverse reactions (*Kuhn et al., 2016*). Drugs from this database with pharmacokinetic profiles that involve entry into the gastrointestinal tract were identified through literature mining and manual curation and indexed by their PubChem CID identifier (*Kim et al., 2016*). Drug use annotations were based on the WHO Anatomical Classification System (*Skrbo et al., 2004*). FooDB (http://foodb.ca/) (*Wishart, 2018*), a database containing raw food component structures, biological interactions and chemical properties was the source of food components linked to PubChem CID identifiers. ClassyFire was used to annotate all xenobiotic and food derived compounds with a shared chemical taxonomy (*Djoumbou Feunang et al., 2016*).

To link microbial enzymes to the set of compounds they metabolize we used KEGGREST (v1.14.1) to retrieve KEGG compound identifiers with links to Enzyme Commission numbers, metabolic

modules and pathways, and presence in either organisms listed as microbial or *Homo sapiens* (*Tenenbaum, 2019*). Enzyme abundance data across human metagenomes were determined based on the total abundance of each enzyme in the healthy participants of the Human Microbiome Project. This data was extracted from the Integrated Microbial Genomes database (*Markowitz et al., 2012*). Enzyme specific dominance scores (ECs$_D$), which is a measure of the number of different species that carry a specific enzyme, were computed based on species-specific enzyme abundance data from healthy individuals from the Integrative Human Microbiome Project (*Proctor et al., 2014*).

## Construction and assessment of the drug-food chemical similarity network

### Chemical similarity calculation

To determine the pairwise chemical substructure similarity between all compounds we used the PubChem 2D molecular fingerprint (*Kim et al., 2016*). The fingerprint is an 881 dimension binary vector in which each bit represents a specific element, functional group, ring system or other discrete chemical entity (*Kim et al., 2016*). Similarity was defined by the Tanimoto coefficient of the molecular fingerprint representations present between two compounds (*Bajusz et al., 2015*).

### Network construction

Similarity scores are percentages of substructure overlap between pairs of compounds and have values between 0 to 1. Similarity scores are filtered such that compound pairs with less than 0.3 substructure similarity were removed. These pairwise similarity scores formed the basis of the undirected chemical similarity network, where nodes represent compounds and edges represent substructure similarity score.

### Network filtering

To cluster compounds in the network based on substructure similarity we used the Walktrap community detection method (*Pons and Latapy, 2006*) implemented in R/igraph v.1.1.1 (*R Development Core Team, 2016*). Within a community, significant similarity scores were defined as those with Z-scores of 1 standard deviation or greater away from the mean (*Baldi and Nasr, 2010*).

### Assessment of compound substructure-based clustering recapitulation of chemical ontology

The MicrobeFDT substructure similarity network is defined by the Tanimoto coefficient of the PubChem 2D molecular fingerprint representations between two compounds (*Kim et al., 2016*; *Bajusz et al., 2015*). To assess how well compound substructure-based clustering recapitulates chemical ontology we compared network features between the MicrobeFDT substructure similarity network and randomized network with the same number of nodes, edges and labels. Each compound label includes a ClassyFire (*Djoumbou Feunang et al., 2016*) schema derived hierarchical set of chemical descriptors. The chemical similarity network was rendered in Cytoscape using NetworkRandomizer (*Martens et al., 2014*). Using the Wilcoxon rank-sum test we compared superclass level chemical descriptors across connected compounds between the real and random network. For the MicrobeFDT network, we also computed the ratio of compounds pairs with matched Superclass annotation to unmatched annotations for all pairs with the same substructure score to assess the relationship between substructure similarity and shared chemical ontology.

### Predicting the probability of association of compound pairs both serving as substrates for an enzyme based on substructure and physiochemical parameters

Each compound pair was assigned one of two labels, associate or non-associated, based on whether both compounds are substrates for the same enzyme (associated) or not (non-associated), given compound-enzyme relationships in the KEGG database (*Kanehisa and Goto, 2000*). The DataWarrior program (*Sander et al., 2015*) was used to identify the following parameter categories for each compound: geometry, functional groups, aromaticity, amino acid composition, polarity and hydrophobicity. In order to translate compound pair substructure and physiochemical parameters into a

probability of overlapping metabolism we used a machine learning approach for generating probability estimates for multi-class classification problems (*Zhang et al., 2013*; *Wu et al., 2004*). Briefly, this approach builds a multi-class prediction model by using pair-wise coupling. We then implemented the prediction model using the probsvm package in R using a one-vs-one decomposition scheme (*Zhang et al., 2013*).

## Assessing toxicity similarity

Toxicity similarity was computed as described by Campillos and colleagues (*Campillos et al., 2008*) with three key steps: (1) extraction and standardization of side effect concepts across drugs of interest; (2) weighting of unique side effect concepts based on frequency of occurrence and correlation with other side effects; and (3) computation of pair-wise toxicity similarity between drugs based on weighted side-effect concept values. Briefly, Campillos et al. curated a dictionary of side-effects based on the Concepts of the Coding Symbols for Thesaurus of Adverse Reaction Terms (COSTART) ontology (*US Food and Drug Administration, 1995*). Side-effect information on therapeutic drug package labels was identified from publicly available sources and searched against this dictionary such that all unique side effect concepts per drug were based on COSTART ontology. For our analysis we used the side effect labels for therapeutic drugs of interest that were extracted from the Medical Dictionary for Regulatory Activities (*Brown et al., 1999*), which is an updated replacement of COSTART, and made publicly available at the download page for SIDER 4.1 which can be found here.

In Campillos et al., each side effect concept was given a rarity score which is the frequency at which it is found across all drug side effect lists. To account for co-dependence between side effects Campillos and colleagues also determined the correlation between all side effects based, using the Tanimoto score between pairs of side effects. This measure is based on how many drugs share a given side effect relative to the number of drugs that have either. The resulting matrix was used as input for the Gerstein-Sonnhammer-Chothia Algorithm (*Gerstein et al., 1994*), to output a score for each concept that down weights concepts that are redundant. We used a publicly available implementation of this algorithm in R available here. Pair-wise toxicity similarity between drugs was computed based on summing the products of weights over all shared side effect concepts between drug pairs. We fit a linear regression to determine whether there is a linear relationship between compound pair substructure similarity and toxicity similarity.

## Taxonomic signatures of microbial enzymes

For each enzyme, we computed an enzyme commission number-specific dominance (ECs$_D$) score. This score is an application of the Simpson's index, which is particularly sensitive to sample evenness (*DeJong, 1975*), and describes the dominance and diversity profile of species carrying the enzyme (*Ofaim et al., 2017*). The taxa-specific enzyme abundance information is based on data collected as a part of the integrative Human Microbiome Project (iHMP) (PRJNA306874) (*Proctor et al., 2014*). ECs$_D$ scores are reported as Simpson index measure (*Simpson, 1949*) subtracted from one, as implemented in the phyloseq R package (*McMurdie and Holmes, 2013*). In this implementation, the Simpson dominance index per enzyme defined by its enzyme commission number (D(EC)) is computed such that $n$ is number of individuals of each species that carry the enzyme and $N$ is the total number of individuals of all species that carry the enzyme (1). For better interpretability, the dominance scores are subtracted from 1 (2).

$$D(EC) = \frac{\sum n(n-1)}{N(N-1)} \tag{1}$$

$$\mathrm{EC_{SD}} = 1 - D(EC) \tag{2}$$

Thus, enzyme functions carried out by small numbers of microbes have values closer to 0 while functions carried out by taxonomically diverse groups have functions closer to 1.

## Altretamine microbiome turnover validation

### Collection and preparation of fecal samples

Fresh fecal samples were provided by three healthy adult men aged 23–30 with no history of antibiotics for 6 months prior to the study. The study was approved by the Albert Einstein College of Medicine Institutional Review Board. Samples were deposited, immediately stored on ice, and processed within 1 hr. One gram stool from each donor was added to 300 mL BHI supplemented with 0.5% glucose (weight/volume) and homogenized. The final fecal slurry was thus comprised of the pooled feces of the three donors at 1% w/v.

### Altretamine metabolism

Fecal slurry cultures were incubated at 37°C in the dark under aerobic conditions. Altretamine stock was prepared in DMSO. Experimental cultures received a final concentration of 100 µM altretamine in DMSO and were prepared in triplicate. Triplicate heat-killed and denatured cultures were autoclaved three times on successive days and also received 100 µM altretamine in DMSO after the third autoclave. Background cultures received fecal slurry and DMSO but no altretamine. To determine matrix effects of altretamine in the media, a sterile media control was amended with 100 µM altretamine in DMSO. Cultures were sampled, immediately snap-frozen in liquid $N_2$ every 24 hr, and stored at −80°C until analysis.

### Altretamine and metabolite quantification

Samples were thawed, centrifuged, and 100 µL aliquots were added to 900 µL 80% methanol. Melamine-triamine-($^{15}N_3$) was used as internal standard. Altretamine and metabolites were identified using LC/MS (Waters Acquity LC system and Waters Xevo TQ MS). Liquid samples were diluted 1:50 in 80% methanol with melamine-triamine-($^{15}N_3$) as internal standard. Each sample was injected 3 times at 5 mL/injection. Separation was performed on an ACE2 C18 column set to 45°C with 0.1% formic acid in 5% methanol (A) and 0.1% formic acid in methanol (B). Elution occurred at 0.35 ml/min with 100% A for 1 min, followed by a 1.5 min linear gradient from 100% A to 95% B, and finally 100% B for 1 min. The voltage was set to 0.044 kV.

## Phylogenetic trees

### N-demethylase phylogenetic tree

N-demethylases from *Pseudomonas putida* CBB5 (*ndmABCD*) (*Summers et al., 2012*) and *Sphingobium* sp. strain YBL2 (*pdmAB*) (*Gu et al., 2013*), both containing a Rieske non-heme iron oxygenase component, catalyze the N-demethylation of phenylurea herbicides and purine alkaloids, respectively; and range in size from 318 to 364 amino acids (*Summers et al., 2012*; *Gu et al., 2013*; *Sharma et al., 2018*). We clustered bacterial N-demethylase sequences described by Summers *et al.,* and Tao *et al.,* as well as protein sequences of >= 200 amino acids in length pulled based on text annotation from the RefSeq database (*Pruitt et al., 2005*) at 95% identity using the UCLUST algorithm (*Edgar, 2010*). The resulting 84 N-demethylase protein sequences served as a protein database which was mapped against the protein calls of healthy adult participants from the Human Microbiome Project (HMP) (PRJNA43021) using the UBLAST algorithm (*Edgar, 2010*) and e-value cutoff of e-40. N-demethylase hits of 200 amino acids or greater formed the basis of a phylogenetic tree which was constructed by aligning the protein sequences using MUSCLE with default parameters (*Edgar, 2004*). Aligned sequences were trimmed at 70% identity and phylogenetic trees were built with PhyML (*Guindon et al., 2010*) with 100 bootstrap replicates, a JTT model of substitution, and otherwise default parameters. The trees were visualized using the packages ggplot2 (*Wickham, 2016*) and phyloseq (*McMurdie and Holmes, 2013*) in R (*R Development Core Team, 2016*). Each branch was colored based on the phylum level classification of the protein, marked by similarity to the experimentally characterized N-demethylase genes *ndmABCD* and *pdmAB* and by the normalized number of total hits found across individuals in the HMP. Black circles indicate bootstrap values of 80/100 or better.

## Bile salt hydrolase phylogenetic tree

We identified bile salt hydrolase protein sequences based on text annotation from the RefSeq database (*Pruitt et al., 2005*) and developed a curated database of protein sequences that were clustered at 95% identity using the UCLUST algorithm (*Edgar, 2010*) resulting in 300 bile salt hydrolase protein sequences with a minimum amino acid length cutoff of 300. Bile salt hydrolase subunits can range in length up to 518 amino acids in the literature (*Breton et al., 2002*; *Bron et al., 2006*; *Schmid and Roth, 1987*). Sequence mapping against the HMP (*Human et al., 2012*), alignment and tree construction were carried out as described for the N-demethylases with the following exception: each branch representing a unique bile salt hydrolase sequence was marked by the presence or absence of reported activity in the literature.

## Acknowledgements

This work was supported by a Peer Reviewed Cancer Research Program Career Development Award from the United States Department of Defense to LK (CA171019). Leah Guthrie was supported in part by the predoctoral Training Program in Cellular and Molecular Biology and Genetics (5T32GM007491-41). Sarah Wolfson was supported in part by NIH/NCI funding (1R01CA222358). The Stable Isotope and Metabolomics Core Facility of the Diabetes Research and Training Center of the Albert Einstein College of Medicine is supported by NIH/NCI funding (P60DK020541). The authors thank the reviewers, members of the Kelly lab and Tyler Grove (Einstein) for helpful suggestions on the work.

## Additional information

### Funding

| Funder | Grant reference number | Author |
| --- | --- | --- |
| National Institutes of Health | Training Program in Cellular and Molecular Biology and Genetics (5T32GM007491-41) | Leah Guthrie |
| National Institutes of Health | 1R01CA222358 | Sarah Wolfson |
| United States Department of Defense | CA171019 | Libusha Kelly |

The funders had no role in study design, data collection and interpretation, or the decision to submit the work for publication.

### Author contributions

Leah Guthrie, Conceptualization, Data curation, Formal analysis, Supervision, Funding acquisition, Visualization, Methodology, Writing—original draft, Writing—review and editing; Sarah Wolfson, Validation, Methodology, Writing—review and editing; Libusha Kelly, Conceptualization, Supervision, Funding acquisition, Writing—original draft, Writing—review and editing

### Author ORCIDs

Leah Guthrie (iD) https://orcid.org/0000-0002-9144-0110
Sarah Wolfson (iD) https://orcid.org/0000-0001-9774-0977
Libusha Kelly (iD) https://orcid.org/0000-0002-7303-1022

### Decision letter and Author response

Decision letter https://doi.org/10.7554/eLife.42866.021
Author response https://doi.org/10.7554/eLife.42866.022

# Additional files

## Supplementary files
• Transparent reporting form
DOI: https://doi.org/10.7554/eLife.42866.019

## Data availability
Data to use or reproduce MicrobeFDT can be found at https://github.com/kellylab/microbeFDT-neo4j (copy archived at https://github.com/elifesciences-publications/microbeFDT-neo4j).

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
