## [Decision Letter]

Thank you for sending your article entitled "The gut chemical landscape predicts microbe-mediated biotransformation of foods and drugs" for peer review at *eLife*. Your article has been evaluated by three peer reviewers, one of whom is a member of our Board of Reviewing Editors, with strong expertise in microbiome research, analytical chemistry, and comparative genomics, respectively, and the evaluation has been overseen by Wendy Garrett as the Senior Editor.

There was a consensus that the identified topic represents a major gap in scientific knowledge and the toolkit established here could provide a valuable resource to the microbiome community. These findings could have broad implications for the field and provide the inspiration and framework for the development of more sophisticated follow-on algorithms. However, the reviewers also raised multiple technical considerations that limited our enthusiasm for publication at this time.

The full reviews are attached below and I have briefly summarized the major concerns here.

1) Lack of computational or experimental validation. The specific examples mentioned as potential future directions do not help that much in evaluating the overall reliability of the predictions made by this approach. The authors should consider whether or not there are suitable gold-standards; for example, the >50 drugs that are already known to be metabolized by human gut bacteria. Alternatively, they could use as positive controls structural analogs like altretamine and melamine, to see if these chemical pairs always result in similar predictions. The authors might also test if their unbiased clustering recapitulates established chemical classes. In addition to these more global analyses of specificity, sensitivity, and precision, we would like to see at least one of the novel predictions validated experimentally (if possible within the time frame).

2) Structural similarity could be misleading. The major caveat to this approach is that overall similarity can be thrown off by differences in parts of the molecules that are irrelevant for the enzymatic activity. This can lead to false positives (compounds with similar overall structure but differences in the relevant sub-structure) and false negatives (compounds with different R groups but the same core chemical motif). While solving this problem may be beyond the scope of this analysis, we would like to see it acknowledged and some analysis to determine how big a problem this is for the current set of predictions.

Numerous research groups work in this area with a higher degree of sophistication, and likely accuracy, than this paper describes (see citations from reviewer 2). Each of the tools above allows compounds to be linked to reactions; none are mentioned in this paper; and none of them rely on simply using a chemical fingerprint as the likelihood that a drug would participate in a chemical reaction. Chemical fingerprints are not intended for asserting that a molecule would be compatible with a chemical reaction. If we understand the algorithm correctly, we don't think it is appropriate for scoring biochemical-reaction promiscuity. We recommend completely overhauling this part of the algorithm.

In the examples that the authors show, leucovorin is strongly clustered with ambenonium which differ in structure. Similarly, metformin and atretamine are very different molecules – even though they are rich in Nitrogen. Furthermore, not every chemical moiety can be undone-just because a molecule has a methyl group doesn't mean that it can be demethylated.

3) Clustering by toxicity could also be misleading. The logic here is not clear, since there could be many reasons for why two molecules show the same toxicity profile – most of which will have nothing to do with the microbiome or with metabolism more broadly. For example, two completely different antidiabetic drugs may have the same side effect, hypoglycemia. A diuretic and a smooth muscle relaxant may have the same side effect, hypotension, etc.

4) ECs are broad groups of enzymes that do not adequately capture substrate preferences. It's unclear to us how the EC analysis would be used in practice. For example, let's consider the case of altretamine and N-demethylases. These enzymes are widespread in bacterial genomes and have broad substrate scope as a family. Which one (if any) can metabolize the drug? Just because an enzyme has an EC annotation of ligase, hydrolase, oxidoreductase, doesn't mean that a molecule with certain functional groups can be a substrate to it. It doesn't even mean that it is correctly annotated.

Reviewer #1:

Guthrie and Kelly present an interesting structure-based analysis of food and drugs with the goal of predicting gut microbial biotransformation. This dataset has great potential as a resource to the community. The major problem is the lack of either computational or experimental validation, making it unclear how reliable the inferences are, and thus limiting their utility. There are also some concerns about the assumptions made by this method that could lead to false positives and false negatives.

Major issues:

1) Lack of validation. While I really enjoyed reading the specific examples mentioned as potential future directions, they do not help that much in evaluating the overall reliability of the predictions made by this approach. The authors should consider whether or not there are suitable gold-standards; for example, the >50 drugs that are already known to be metabolized by human gut bacteria. Alternatively, they could use as positive controls structural analogs like altretamine and melamine, to see if these chemical pairs always result in similar predictions. The authors might also test if their unbiased clustering recapitulates established chemical classes. In addition to these more global analyses of specificity, sensitivity, and precision, I would like to see at least one of the novel predictions validated experimentally.

2) Structural similarity could be misleading. The major caveat to this approach is that overall similarity can be thrown off by differences in parts of the molecules that are irrelevant for the enzymatic activity. This can lead to false positives (compounds with similar overall structure but differences in the relevant sub-structure) and false negatives (compounds with different R groups but the same core chemical motif). While I realize that solving this problem is beyond the scope of this analysis, I would like to see it acknowledged and some analysis to determine how big a problem this is for the current set of predictions.

3) ECs are broad groups of enzymes that do not adequately capture substrate preferences. It's unclear to me how the EC analysis would be used in practice. For example, let's consider the case of altretamine and N-demethylases. These enzymes are widespread in bacterial genomes and have broad substrate scope as a family. Which one (if any) can metabolize the drug?

4) The section on antibiotics (subsection “MicrobeFDT identifies food derived compounds and non-antibiotic therapeutic drugs with putative antimicrobial properties”) was interesting, but doesn't fit the scope of this paper. Need to either remove this section or reframe the goal of this study. I'd vote for the former, since it is currently lacking in novelty or experimental validation.

Reviewer #2:

The manuscript, "The gut chemical landscape predicts microbe-mediated biotransformation of foods and drugs", herein referred to as "this paper", describes a creative computational link between drugs, metabolites, genes and microbes. The narrative on the importance of linking the compounds we eat whether drugs or from food with the reactions in the gut performed by microbes is almost worth publishing on its own. It was enjoyable to read such a well thought out and presented essay on this topic. In regards to the actual research described in the paper, I have two major concerns: the validity of their major claims and using chemical fingerprint as a proxy for reactivity given alternative approaches.

To see if other e*Life* manuscripts contained more or less validation of computational methods, I searched *eLife* for "chemical fingerprint". There were 43 results. Of these, the following three are in the same general area as this paper.

– Prediction of enzymatic pathways by integrative pathway mapping: https://doi.org/10.7554/eLife.31097

– Systematic integration of biomedical knowledge prioritizes drugs for repurposing: https://doi.org/10.7554/eLife.26726

– Digitizing mass spectrometry data to explore the chemical diversity and distribution of marine cyanobacteria and algae: https://doi.org/10.7554/eLife.24214

In these three prior manuscripts, the amount of validation varies quite a lot. "Prediction of enzymatic…" includes enzymology, crystallography, and metabolomics to validate their method for a specific pathway; "Systematic integration…" uses only out-of-sample validation set; and "Digitizing mass spectrometry…" uses the identification of a single novel compound to suggest validation of the overall method. My initial reaction to this paper was that its an important application of a creative idea, but the degree to which the network they've built is validate is insufficient. However, in comparison to the prior papers in *eLife*, I think the amount of validation in this paper is only slightly lower than what is already in the journal. Testing the predictions of their network would be a significant endeavor.

Nevertheless, this paper makes the following claims:

"Together, our resource identifies novel gut microbiome-mediated metabolic activity and associated adverse responses that can be used to identify targets for experimental validation and to generate new hypotheses about microbe-drug-diet interactions in human health and disease. We demonstrate the utility of this resource with the following three novel insights into microbial drug metabolism and human health."

And the following speculations:

"We propose uninvestigated microbiota mediated metabolisms that may drive toxicity of therapeutic drugs, we highlight non-antibiotic compounds that may have antimicrobial properties, and we identify drug-food interactions with microbial enzymes that may influence drug efficacy and microbiome function."

For all claims made in the paper, my belief is that a validated approach should be used. Therefore, I recommend the authors validate their approach prior to making such claims. In the scope of this journal, speculation and suggestion is ok as long as it is worded appropriately.

I'm worried about relying on a chemical fingerprint to assert that a drug would react in place of a metabolite in a known biochemical reaction. Numerous research groups work in this area with a higher degree of sophistication, and likely accuracy, than this paper describes. I am aware of the following works, but there are likely many more.

– in vivo/In Silico Metabolites Database (IIMDB)

– RetroPath and RetroRules (http://www.jfaulon.com/)

– MINEs: open access databases of computationally predicted enzyme promiscuity products for untargeted metabolomics

– MAGI: (https://magi.nersc.gov)

– ATLAS of Biochemistry: A Repository of All Possible Biochemical Reactions for Synthetic Biology and Metabolic Engineering Studies

– Nontargeted in vitro metabolomics for high-throughput identification of novel enzymes in *Escherichia coli*. Nature Methods (2016)

Each of the tools above allows compounds to be linked to reactions; none are mentioned in this paper; and none of them rely on simply using a chemical fingerprint as the likelihood that a drug would participate in a chemical reaction. Perhaps the most directly applicable to this paper would be RetroPath/RetroRules. It uses a considerably more elegant means to assert that a drug would participate in a reaction. Chemical fingerprints are not intended for asserting that a molecule would be compatible with a chemical reaction. If I understand your algorithm correctly, I don't think it is appropriate for scoring biochemical-reaction promiscuity. I recommend completely overhauling this part of the algorithm.

Reviewer #3:

In the submitted manuscript, Guthrie and Kelly propose a resource (MicrobeFDT) that constructs interacting networks of drugs, endogenous, and dietary molecules based on their structural similarity, known toxicities, and possible metabolizing enzymes.

The manuscript was written in a very convoluted manner, with intertwined logic, design, and results (and even literature references), making it difficult to follow. There is very limited detail about the approach itself and actual analyses. The authors describe three main tasks:

1) Define the chemical space of xenobiotics that the human microbiome is exposed to. They use similarity in chemical substructure to achieve this goal, based on previously developed algorithms and metrics. In the one example that the authors show, leucovorin is strongly clustered with ambenonium – which is clearly a problem! Similarly, metformin and atretamine are very different molecules – even though they are rich in Nitrogen!

2) Cluster molecules based on their reported toxicities. The logic here is not clear, since there could be many reasons for why two molecules show the same toxicity profile – most of which will have nothing to do with the microbiome. For example, two completely different antidiabetic drugs may have the same side effect, hypoglycemia. A diuretic and a smooth muscle relaxant may have the same side effect, hypotension, etc.

3) Identify biochemically relevant and functionally plausible microbe-compound interactions. This section was not clear at all to the reviewer. Even the essence for how microbial enzymes are linked to the network of molecules is not described clearly. In one section, the authors mention: "structure-activity filtering, the user establishes the search target based on prior knowledge of structure-activity relationships (for example, a compound with a methyl group that is susceptible to microbial methylases)". Why would a methylated compound be susceptible to methylases? Maybe it is a typo and the authors meant demethylases? Even in this case, the logic is very unclear. Not every chemical moiety can be undone, and just because a molecule has a methyl group doesn't mean that it can be demethylated. Also, there is a fundamental misunderstanding here for the basics of biochemistry. Just because an enzyme has an EC annotation of ligase, hydrolase, oxidoreductase, doesn't mean that a molecule with certain functional groups can be a substrate to it. It doesn't even mean that it is correctly annotated. Basing conclusions on these general annotations is meaningless.

Overall, the reviewer does not see the value of the tool developed here, based on fundamental issues with the logic of the design itself, as well as a lack of clarity in the description of the method and analysis performed. The examples listed in the text are either obvious (e.g., glucoronidated molecules will probably get deglucoronidated) or completely speculative (steroids are antibiotics), despite being "supported" by cherry-picked citations.

---

## [Author Response]

Thank you for sending your article entitled "The gut chemical landscape predicts microbe-mediated biotransformation of foods and drugs" for peer review at eLife. Your article has been evaluated by three peer reviewers, one of whom is a member of our Board of Reviewing Editors, with strong expertise in microbiome research, analytical chemistry, and comparative genomics, respectively, and the evaluation has been overseen by Wendy Garrett as the Senior Editor.There was a consensus that the identified topic represents a major gap in scientific knowledge and the toolkit established here could provide a valuable resource to the microbiome community. These findings could have broad implications for the field and provide the inspiration and framework for the development of more sophisticated follow-on algorithms. However, the reviewers also raised multiple technical considerations that limited our enthusiasm for publication at this time.The full reviews are attached below and I have briefly summarized the major concerns here.1) Lack of computational or experimental validation. The specific examples mentioned as potential future directions do not help that much in evaluating the overall reliability of the predictions made by this approach. The authors should consider whether or not there are suitable gold-standards; for example, the >50 drugs that are already known to be metabolized by human gut bacteria. Alternatively, they could use as positive controls structural analogs like altretamine and melamine, to see if these chemical pairs always result in similar predictions. The authors might also test if their unbiased clustering recapitulates established chemical classes. In addition to these more global analyses of specificity, sensitivity, and precision, we would like to see at least one of the novel predictions validated experimentally (if possible within the time frame).

Computational validation

We agree that computational validation is vital to demonstrate the value of the MicrobeFDT resource. To computationally validate MicrobeFDT, we 1) demonstrate that our structural similarity metric can identify a) shared enzyme metabolism and b) shared drug toxicity; 2) show that our clustering recapitulates established chemical classes; and 3) perform an out-of-sample validation to demonstrate high substructure overlap correlates with enzyme specificity.

1a) Assessing the correlation between compound structural similarity and shared enzyme metabolism.

We characterize how well the structural similarity metric in MicrobeFDT enables identification of compounds that share metabolism by an enzyme and demonstrate that structural similarity can be used as filter to identify shared metabolism.

“To validate that our network can identify shared metabolism, we developed an *in silico* prediction model to assign a probability of shared metabolism between compounds based on substructure overlap and the following physiochemical categories: geometry, functional groups, amino acid composition, polarity and hydrophobicity. We find that the probability of compound-pairs sharing an enzyme based on substructure and physiochemical parameters increase as the substructure overlap score between compound pairs increases (Figure 2). Weighting compound pair chemical similarity relationships based solely on substructure similarity is thus a reasonable filtering step to identify compounds that may share metabolism.

“Predicting the probability of association of compound pairs both serving as substrates for an enzyme based on substructure and physiochemical parameters.

Each compound pair was assigned one of two labels, associate or non-associated, based on whether both compounds are substrates for the same enzyme (associated) or not (non-associated), given compound-enzyme relationships in the KEGG database^102^. The DataWarrior program^103^ was used to identify the following parameter categories for each compound: geometry, functional groups, aromaticity, amino acid composition, polarity and hydrophobicity. In order to translate compound pair substructure and physiochemical parameters into a probability of overlapping metabolism we used a machine learning approach for generating probability estimates for multi-class classification problems^104,105^. Briefly, this approach builds a multi-class prediction model by using pair-wise coupling. We then implemented the prediction model using the probsvm package in R using a one-vs-one decomposition scheme^104^”

1b) Assessing the correlation between structural similarity and toxicity similarity.

To evaluate whether our network also recapitulates shared drug toxicity found in earlier studies (Campillos et al., 2008) we fit a linear regression and computed the effect size to assess the association between substructure similarity and toxicity similarity for therapeutic drugs in our network.

“Previous studies have found that structural similarity predicts both toxicity and drug target similarity^44^. To evaluate whether our network also recapitulates shared drug toxicity we fit a linear regression and computed the effect size to assess the association between substructure similarity and toxicity similarity for therapeutic drugs in our network. We find that structural similarity moderately positively predicts toxicity similarity for therapeutic drug pairs linked by structural similarity overall in the network; the signal is stronger at higher structural similarity (r = 0.03116, p < 2.2e-16) (Figure 4).”

“Pair-wise toxicity similarity between drugs was computed based on summing the products of weights over all shared side effect concepts between drug pairs. We fit a linear regression to determine whether there is a linear relationship between compound pair substructure similarity and toxicity similarity.”

2) Clustering validation

We demonstrate that our clustering recapitulates established chemical classes. For clusters of compounds in the chemical similarity network we evaluated how often they recapitulate structure-based chemical taxonomy as defined by the ClassyFire resource, a comprehensive chemical classification schema. We found that substructure-based compound clustering, significantly groups compounds within a ClassyFire superclass based on a comparison of the MicrobeFDT network with a randomized network with the same number of nodes and edges.

“Finally, we evaluated how well our compound clustering recapitulates structure-based chemical taxonomy as defined by the ClassyFire^45^ resource, a comprehensive chemical classification schema, at the level of superclass taxonomy. We found that substructure-based compound clustering, significantly groups compounds within a ClassyFire superclass based on a comparison of the MicrobeFDT network with a randomized network with the same number of nodes and edges (p < 8.06 x 10-15, Wilcoxon rank-sum test). Compound-pairs at higher substructure similarity share Superclass membership at higher substructure values and at a greater frequency than randomized pairs, indicating that the MicrobeFDT substructure similarity metric can capture established chemical classifications (Figure 5).”

“Assessment of compound substructure-based clustering recapitulation of chemical ontology.

The MicrobeFDT substructure similarity network is defined by the Tanimoto coefficient of the PubChem 2D molecular fingerprint representations between two compounds^3839^. To assess how well compound substructure-based clustering recapitulates chemical ontology we compared network features between the MicrobeFDT substructure similarity network and randomized network with the same number of nodes, edges and labels. Each compound label includes a ClassyFire schema derived hierarchical set of chemical descriptors. The chemical similarity network was rendered in Cytoscape using NetworkRandomizer^1^. Using the Wilcoxon rank-sum test we compared superclass level chemical descriptors across connected compounds between the real and random network. For the MicrobeFDT network, we also computed the ratio of compounds pairs with matched Superclass annotation to unmatched annotations for all pairs with the same substructure score to assess the relationship between substructure similarity and shared chemical ontology.”

3) Out of sample validation of compound-enzyme interactions.

We do an out-of-sample validation by assessing substructure overlap of compounds in the network with digoxin, a cardiac glycoside drug and a substrate of the Cgr2 enzyme. This microbial enzyme has the most complete drug substrate profile to our knowledge and we therefore selected it to demonstrate the utility of a substructure similarity screen to identify compounds that might interact with the same enzyme.

“As an example of how the network can reveal shared metabolism we selected compounds in the network with substructure overlap with digoxin, a cardiac glycoside. Reduction of digoxin by a human microbiome reductase inactivates the drug, contributing to poor bioavailability in some individuals^5,9,42^. Koppel et al., biochemically characterized the capacity of a single flavin- and [4Fe-4S] cluster-dependent reductase, *cgr2*, to reduce various substrates with a range of substructure similarity to digoxin^43^. We identified the substructure overlap between digoxin and compounds in the Koppel et al. study that were evaluated as substrates of Cgr2 enzyme. Among the biochemically assayed compounds^43^ that are present in the MicrobeFDT network, compounds with substructure similarity scores greater than 0.8 are also substrates for Cgr2. This assessment suggests that for the cgr enzyme substructure based clustering can distinguish experimentally characterized substrates from non-substrates (Figure 3).”

Together, these additional analyses and figures validate the computational methods used to build MicrobeFDT.

Experimental validation of the MicrobeFDT resource.

As noted by the reviewers, experimental validation is a major undertaking. We agree with the reviewers that experimental validation is also necessary to demonstrate that MicrobeFDT can identify novel, potentially biologically and clinically relevant microbiome/compound interactions. Sarah Wolfson, a postdoc in the Kelly lab, was instrumental in these experiments and we have added her as a coauthor to the manuscript. A combination of features in MicrobeFDT indicates that altretamine, an ovarian cancer drug, may be demethylated by N-demethylase enzymes in the microbiome. This prediction comes from structural overlap, toxicity similarity, and inferred enzyme/compound interactions from the network. We validated microbiome metabolism of altretamine by incubating altretamine in a pooled fecal slurry generated from three individuals and monitored altretamine and potential metabolites using LC-MS/MS. We found that a metabolite that is structurally identical to pentamethylmelamine, a demethylated altretamine metabolite, increases in active fecal microcosms over 48 hours. We thus experimentally validate a novel microbiome/drug interaction predicted by MicrobeFDT.

“A first step in validating this hypothesis is to demonstrate that the gut microbiome can demethylate altretamine. We incubated altretamine in a pooled fecal slurry generated from three individuals and monitored altretamine and potential metabolites using LC-MS. We controlled for the formation of spontaneous N-demethylation of altretamine, which has been reported in the literature^58^, and found that a metabolite that is structurally identical to pentamethylmelamine, a demethylated altretamine metabolite, increases in active fecal microcosms over 48 hours (Figure 9). In active fecal biotic conditions the metabolite continually increased between time 0 and 48 hours. Killed controls demonstrated an increase in metabolite between 0 and 24 hours, though to a lesser extent than in active fecal microcosms. Notably there was little metabolite formation after 24 hours, indicating that in addition to abiotic N-demethylation, active gut microbes demethylate a substantial portion of altretamine to the putative metabolite pentamethylmelamine.”

“Altretamine microbiome turnover validation

Collection and preparation of fecal samples:

Fresh fecal samples were provided by 3 healthy adult men aged 23-30 with no history of antibiotics for 6 months prior to the study. […] Elution occurred at 0.35 ml/min with 100% A for 1 minute, followed by a 1.5 minute linear gradient from 100% A to 95% B, and finally 100% B for 1 minute. The voltage was set to 0.044 kV.”

2) Structural similarity could be misleading. The major caveat to this approach is that overall similarity can be thrown off by differences in parts of the molecules that are irrelevant for the enzymatic activity. This can lead to false positives (compounds with similar overall structure but differences in the relevant sub-structure) and false negatives (compounds with different R groups but the same core chemical motif). While solving this problem may be beyond the scope of this analysis, we would like to see it acknowledged and some analysis to determine how big a problem this is for the current set of predictions.Numerous research groups work in this area with a higher degree of sophistication, and likely accuracy, than this paper describes (see citations from reviewer 2). Each of the tools above allows compounds to be linked to reactions; none are mentioned in this paper; and none of them rely on simply using a chemical fingerprint as the likelihood that a drug would participate in a chemical reaction. Chemical fingerprints are not intended for asserting that a molecule would be compatible with a chemical reaction. If we understand the algorithm correctly, we don't think it is appropriate for scoring biochemical-reaction promiscuity. We recommend completely overhauling this part of the algorithm.In the examples that the authors show, leucovorin is strongly clustered with ambenonium which differ in structure. Similarly, metformin and atretamine are very different molecules – even though they are rich in Nitrogen. Furthermore, not every chemical moiety can be undone-just because a molecule has a methyl group doesn't mean that it can be demethylated.

The reviewers pointed out several additional major issues with our work and we address below 1) how our compound-enzyme interactions are defined and 2) our decision to use a chemical fingerprint to assess structural similarity among compounds. We agree with the reviewers that the fingerprint is not suitable for scoring biochemical-reaction promiscuity.

1) Compound-enzyme link clarification

We regret that it was unclear that the network does not predict the links between compounds and enzymes; rather these links in the network are derived from published annotations in the Kyoto Encyclopedia of Genes and Genomes (KEGG).

We indicate in the text compound and enzyme links are generated from available resources as follows: “To link microbial enzymes to the set of compounds they metabolize we used KEGGREST (v1.14.1) to retrieve KEGG compound identifiers with links to Enzyme Commission numbers, metabolic modules and pathways, and presence in either organisms listed as microbial or *Homo sapiens^97^*.”

We agree with the comment “not every chemical moiety can be undone” and we agree that the fingerprint is not suitable to predict biochemical-reaction promiscuity. These two comments underlie why we did not attempt to predict novel compound/enzyme associations in MicrobeFDT but rather developed the resource to shrink the search space of potential compound/enzyme interactions for users.

For the altretamine example, we hypothesize that N-demethylase/altretamine interactions contribute to drug toxicity based on the observations that 1) altretamine clusters structurally with melamine; 2) altretamine and melamine have similar toxicities in MicrobeFDT; and 3) altretamine is linked in the network to microbial N-demethylases.

“To provide a practical example of using multiple features of MicrobeFDT to identify uninvestigated microbiota-driven drug toxicity, we searched the network for compounds with high structural and toxicity similarity. […] We hypothesized that gut microbial N-demethylases may partially or completely N-demethylate altretamine, converting it into metabolites that contribute to patient toxicity.”

Importantly, the network does not predict demethylation. For all compounds in the network enzymatic activity is described only by existing annotations in KEGG. Rather the network reveals structural and toxicity similarity of these two compounds and we use these features to hypothesize a mechanism by which N-demethylases may act on the compound to produce toxic intermediates. The power of MicrobeFDT is that it connects structural similarity with biochemistry, toxicity, and microbial ecology and we have corrected our presentation of the work in the revision to clarify that structural similarity is not intended to be the sole basis of a prediction.

“The chemical space of the human gastrointestinal tract ecosystem is shaped by host dietary intake, xenobiotic exposure, and host and gut microbiome derived products. […] We used a combination of network features to successfully predict the novel microbial metabolism of the cancer drug altretamine.”

2) The strengths and limitations of the chemical fingerprint as the foundation for the structural similarity network.

We agree that overall structural similarity, as defined by the chemical fingerprint approach we take, does not prioritize fragment substructures based on susceptibility to enzyme mediated chemical modifications or take into account the active moieties of a compound. We have included extensive revisions to the manuscript to address these points. (New figures 2, 3, 4, 5, 9; text revisions as described above)

The reviewers also point to the role of enzyme promiscuity in shaping the probability that two chemically overlapping compounds will be processed by the same enzyme. To include enzyme promiscuity in MicrobeFDT we will discuss extracting data from resources like RetroRules^9^ (mentioned by the reviewers), which uses SMARTS strings to define reaction rules, and/or the Promis server. The Promis server provides a measure of enzyme multifunctionality that could further support a user’s ranking of hypothesized compound-enzyme interactions. We agree with the reviewers that it is important to describe the many issues with defining structural similarity between molecules. We describe these alternative chemical reaction prediction and discuss possibilities for improving the inference of biochemical interactions between molecules and enzymes in future iterations of the resource.

“A combination of compound, host, and microbiome features influence potential microbial metabolism. Examining these features individually cannot reliably infer clinical phenotypes associated with microbiome/compound interactions. Two molecules may have the same toxicity profile but very different biochemistry, for example. Automated enzyme annotation may be incorrect, and compound structural similarity is often insufficient to predict substrate preferences. Finally, enzymes that carry out a reaction associated with a patient phenotype may be unevenly distributed across microbes and across human microbiomes. MicrobeFDT is designed to overcome some of these limitations by enabling a more holistic analysis of toxicity, structure, metabolism and ecology.”

“MicrobeFDT does not predict substrate specificity for microbiome enzymes; available data and methods are not sufficient to achieve this goal. Enzyme promiscuity also shapes the probability that two chemically overlapping compounds will be processed by the same enzyme. A future improvement to our resource could extract data from resources like RetroRules^91^, which uses SMARTS strings to define reaction rules, or utilizing the Promis server measure of enzyme multifunctionality^92^ to further support a user’s ranking of hypothesized compound-enzyme interactions.”

3) Clustering by toxicity could also be misleading. The logic here is not clear, since there could be many reasons for why two molecules show the same toxicity profile – most of which will have nothing to do with the microbiome or with metabolism more broadly. For example, two completely different antidiabetic drugs may have the same side effect, hypoglycemia. A diuretic and a smooth muscle relaxant may have the same side effect, hypotension, etc.

We agree that some side effects are common across drugs with different metabolic pathways. For this reason, we used a measure of toxicity similarity that weights side effects based on frequency, correlation and co-dependence across all compounds in our dataset; this approach has been experimentally tested to predict previously unknown molecular interactions based on side-effect data^2^. We also agree that it is important to address the limitations of clustering compounds by toxicity alone. Again we regret that we were unclear that toxicity profiling is only one part of the MicrobeFDT resource and it is not intended to be used as the sole feature on which to make a prediction of drug/microbiome interactions. Our changes that validate toxicity clustering, and the importance of using multiple features in MicrobeFDT to make a prediction, are detailed above.

4) ECs are broad groups of enzymes that do not adequately capture substrate preferences. It's unclear to us how the EC analysis would be used in practice. For example, let's consider the case of altretamine and N-demethylases. These enzymes are widespread in bacterial genomes and have broad substrate scope as a family. Which one (if any) can metabolize the drug? Just because an enzyme has an EC annotation of ligase, hydrolase, oxidoreductase, doesn't mean that a molecule with certain functional groups can be a substrate to it. It doesn't even mean that it is correctly annotated.

We agree that the EC number does not capture substrate specificity and we agree that microbial enzymes in the gut are widespread and can have broad substrate scope. We clarify that MicrobeFDT does not predict substrate specificity and we discuss why it is useful to assess how widespread the distribution of a function of interest in the gut.

“An important step towards characterizing the role of the gut microbiome in shaping individual responses to foods and drugs is identifying how gut microbiome metabolism varies from compound to compound and how this metabolism relates to inter-personal variation in diet or drug responses to specific compounds. […] The ECs_D_ score is a readout of potential substrate metabolism at the community level that can be linked to inter-personal variation in gut function and phenotypic outcomes.”

“MicrobeFDT does not predict substrate specificity for microbiome enzymes; available data and methods are not sufficient to achieve this goal. Enzyme promiscuity also shapes the probability that two chemically overlapping compounds will be processed by the same enzyme. A future improvement to our resource could extract data from resources like RetroRules^91^, which uses SMARTS strings to define reaction rules, or utilizing the Promis server measure of enzyme multifunctionality ^92^ to further support a user’s ranking of hypothesized compound-enzyme interactions.”

We do not attempt to predict which N-demethylases act on altretamine. We agree with the reviewers that there are many N-demethylases in the human microbiome (Figure 4—figure supplement 1) and it is not possible to predict computationally which specific enzymes are interacting with a compound of interest. As noted by the reviewers, there are many other widely distributed enzymes with broad substrate scope in the microbiome. To discuss another example, we do not predict all the carbohydrates that a particular carbohydrate active enzyme in the microbiome, such as a beta glucuronidase, acts on. We also agree, and will note in the revision, that there are likely misannotations in the databases we are using that incorrectly ascribe functions to particular enzymes.

“Automated enzyme annotation may be incorrect, and compound structural similarity is often insufficient to predict substrate preferences.”

We hope it is now clear in the text that MicrobeFDT does not predict substrate specificity for all microbiome enzymes; both our methods and the available data make it impossible to achieve this goal currently. But as also noted by the reviewers, our resource is a vital, missing first step towards this goal.

The examples of potential food/drug interactions with SN-38G metabolism by the gut and potential antimicrobial effects of bile salt-like compounds are biologically reasonable examples of novel hypotheses of microbiome/food/drug interactions that MicrobeFDT can generate. These vignettes are intended as starting points for experimental characterization but they also indicate ways for a user to interact with MicrobeFDT to generate hypotheses with the starting point of a specific compound (like SN-38G) or with a very broad function (like antimicrobial activity). We therefore request that we be allowed to leave these examples as part of the manuscript.

We hope that we have sufficiently addressed reviewer 2’s major concerns regarding “The validity of their major claims and using chemical fingerprint as a proxy for reactivity given alternative approaches.” in the revisions described above.

The revisions and responses above also address reviewer 3’s major concerns about how we 1) define the chemical space of xenobiotics that the human microbiome is exposed to; 2) cluster molecules based on their reported toxicities; and 3) identify biochemically relevant and functionally plausible microbe-compound interactions. We hope that computational and experimental validation of MicrobeFDT moves reviewer 3 to now see the value in our resource.